# Sleep-like cortical OFF-periods disrupt causality and complexity in the brain of unresponsive wakefulness syndrome patients

M. Rosanova [1,2,3], M. Fecchio [1], S. Casarotto [1,4], S. Sarasso [1], A.G. Casali[5], A. Pigorini[1], A. Comanducci[1], F. Seregni[6], G. Devalle[4], G. Citerio [7], O. Bodart [8], M. Boly[9,10], O. Gosseries[8], S. Laureys[8] & M. Massimini[1,4]

Unresponsive wakefulness syndrome (UWS) patients may retain intact portions of the thalamocortical system that are spontaneously active and reactive to sensory stimuli but fail to engage in complex causal interactions, resulting in loss of consciousness. Here, we show that loss of brain complexity after severe injuries is due to a pathological tendency of cortical circuits to fall into silence (OFF-period) upon receiving an input, a behavior typically observed during sleep. Spectral and phase domain analysis of EEG responses to transcranial magnetic stimulation reveals the occurrence of OFF-periods in the cortex of UWS patients ($N = 16$); these events never occur in healthy awake individuals ($N = 20$) but are similar to those detected in healthy sleeping subjects ($N = 8$). Crucially, OFF-periods impair local causal interactions, and prevent the build-up of global complexity in UWS. Our findings link potentially reversible local events to global brain dynamics that are relevant for pathological loss and recovery of consciousness.

[1] Department of Biomedical and Clinical Sciences "L. Sacco", University of Milan, Milan 20157, Italy. [2] Fondazione Europea per la Ricerca Biomedica Onlus, Milan 20063, Italy. [3] Neurointensive Care Unit, ASTT Grande Ospedale Metropolitano Niguarda, Milan 20162, Italy. [4] IRCCS Fondazione Don Gnocchi, Milan 20149, Italy. [5] Instituto de Ciência e Tecnologia, Universidade Federal de São Paulo, Sao Jose dos Campos 12231-280, Brazil. [6] Department of Paediatrics, Cambridge University Hospital NHS Foundation Trust, Cambridge CB2 0QQ, UK. [7] Scuola di Medicina e Chirurgia, University of Milan Bicocca, Milan 20126, Italy. [8] GIGA-consciousness, Coma Science Group, University and University Hospital of Liège, Liège 4000, Belgium. [9] Department of Neurology, University of Wisconsin, Madison, WI 53705, USA. [10] Department of Psychiatry, University of Wisconsin, Madison, WI 53719, USA. These authors equally contributed: Rosanova M., Fecchio M. Correspondence and requests for materials should be addressed to M.M. (email: marcello.massimini@unimi.it)

Patients diagnosed with unresponsive wakefulness syndrome (UWS), previously known as vegetative state[1], can open their eyes, recover sleep–wake cycles, but do not show behavioral signs of consciousness[2]. Despite behavioral unresponsiveness, many of these patients retain large parts of the thalamocortical system that are structurally intact, spontaneously active[3,4] as well as reactive to sensory stimuli, though cortical responses tend not to propagate beyond primary areas[3,5,6]. Preserved cortical reactivity in UWS patients can be directly demonstrated by measuring the electroencephalographic response to transcranial magnetic stimulation (TMS/EEG); apart from severe post-anoxic patients, TMS always elicits significant cortical responses in UWS patients. In a minority of such patients, EEG responses to TMS are similar to those observed in conscious subjects, suggesting that they may retain a covert capacity for consciousness. However, in most cases, the EEG response to TMS is simple and stereotypical, as assessed by the perturbational complexity index (PCI): in these patients, identified as "low-complexity" UWS, TMS elicits a strong initial activation, which fails to evolve into complex patterns of interactions[7]. In summary, in many UWS patients cortical circuits seem to be active, reactive but blocked in a pathological low-complexity state.

Non-Rapid Eye Movement (NREM) sleep is a physiological condition in which thalamocortical circuits are structurally intact, functionally active and reactive, yet unable to engage in long-range, complex responses[8,9]. Recent studies employing intracortical stimulation and simultaneous local field potential recordings in humans suggest that the mechanism responsible for this impairment in NREM sleep is the tendency of cortical neurons to fall into a period of suppressed firing (OFF-period) after a transient increase in activity[10,11]. This intrinsic propensity of cortical neurons to fall into OFF-periods has been thoroughly studied in the realm of sleep physiology across species and models and is often referred to as cortical bistability[12,13]. In silico, in vitro as well as in vivo animal models suggests that cortical bistability is due to adaptation mechanisms, such as activity-dependent $K^+$ currents[14,15] as well as active inhibition[16,17]. Crucially, intracranial measurements in sleeping humans show that, due to cortical bistability, neurons react briefly to incoming signals and then fall into an OFF-period, which rapidly disrupts the cause-effects chain triggered by the initial input. Thus, in physiological sleep a simple mechanism leads to a breakdown of deterministic responses and prevents the emergence of sustained, complex patterns of interaction, despite preserved activity and reactivity.

Can a pathological form of bistability play a role also in the residual cortex of low-complexity UWS patients? Asking this question is relevant for at least two reasons. First, cortical bistability and OFF-periods represent a basic default mode of cortical activity[18], which can be engendered by physiological changes as well as by pathological alterations, such as shifts of the inhibition/excitation balance[19] or white matter lesions[20]. Second, OFF-periods can disrupt complex cortico-cortical interactions, but are in principle reversible.

Here, we specifically test the following hypotheses: (i) pathological sleep-like OFF-periods occur in the cortex of awake UWS patients and (ii) this mechanism is responsible for the collapse of causality and overall brain complexity associated with loss of consciousness following brain injury. To do so, we analyzed TMS-evoked EEG potentials recorded in low-complexity UWS patients with the same analysis previously used on intracranially-evoked local field potentials during sleep[10]. First, we show that in UWS patients with their eyes open, the EEG response to TMS in anatomically preserved cortical areas matches the electrophysiological criteria for the detection of an OFF-period, as assessed during NREM sleep, i.e. the presence of a simple positive-negative wave, associated with a suppression of high-frequency activity. Next, we demonstrate that OFF-periods rapidly disrupt the local causal effects of TMS (as indexed by phase-locking measures) and in turn, the emergence of global complex cortico-cortical interactions (as indexed by PCI).

## Results

**Measurements in UWS, sleep and wakefulness**. We analyzed 72 TMS/EEG measurements performed in 16 awake UWS patients, as assessed by the Coma Recovery Scale-Revised (CRS-R[21]), and 20 healthy subjects during wakefulness and NREM sleep, while stimulating both frontal and parietal cortex. Specifically, we assessed (1) the occurrence of TMS-evoked slow waves (<4 Hz) associated with the presence of cortical OFF-periods, i.e. significant high frequency (>20 Hz) suppression of EEG power compared to baseline[22–24], (2) the impact of the OFF-periods on local causal interactions quantified by means of broadband (>8 Hz) phase-locking factor (PLF), (3) the consequences of the OFF-period on the build-up of complex global interactions as indexed by the time course of PCI. For a detailed description of the experimental and analytical procedures, see the Methods section and Supplementary Fig. 1.

**TMS reveals sleep-like cortical OFF-periods in UWS patients**. TMS-evoked EEG potentials recorded in UWS patients consisted of a slow wave, which was associated with an initial activation rapidly followed by a significant high frequency (>20 Hz) suppression of EEG power (HFp) starting at around $103 \pm 9$ ms (mean ± SEM; Fig. 1b and Supplementary Fig. 2B). This pattern of local reactivity, matching the criteria for an OFF-period[22–24], was observed in all stimulated areas (both frontal and parietal bilaterally; Fig. 2a) in each of the 16 UWS patients. As shown in Fig. 2b TMS-evoked slow waves and OFF-periods could be detected irrespective of the presence/absence of spontaneous slow waves in the ongoing pre-stimulus activity (Fig. 2b–d).

The responses found in UWS patients differed markedly from awake healthy subjects stimulated over the same areas (Fig. 1a and Supplementary Fig. 2A); in this latter case, evoked slow waves were absent, low-frequency (<4 Hz) EEG amplitude (max SWa, see Methods section) was significantly lower (Wilcoxon ranksum test, $P = 0.014$ and $P = 0.010$ for parietal and frontal stimulation, respectively; Fig. 1d and Supplementary Fig. 2C, top panel; Table 1 and Supplementary Table 1) and the suppression of high-frequency power was never observed (Fig. 1d and Supplementary Fig. 2C, middle panel). Conversely, the UWS response was similar to the one found in healthy subjects during NREM (Fig. 1c), where TMS evoked a slow wave with a comparable level of low-frequency EEG amplitude (Wilcoxon ranksum test, $P = 0.357$; Fig. 1d, top panel; Table 1) associated with a significant high-frequency suppression (Wilcoxon ranksum test, $P = 0.999$; Fig. 1d, middle panel; Table 1) starting at around $127 \pm 11$ ms.

**Cortical OFF-periods disrupt local causality in UWS patients**. The duration of the causal effects of TMS on local cortical activity, as assessed by the PLF, was short-lived in UWS patients. Indeed, the latest significant PLF value (max PLFt, see Methods section) occurred at $167 \pm 21$ ms (mean ± SEM) when stimulating parietal cortex and at $188 \pm 18$ ms when stimulating frontal cortex (Fig. 1d, Supplementary Fig. 2C). These values roughly corresponded to the timing of the maximum of high frequency (>20 Hz) suppression (max SHFt) and were similar to the max PLFt of healthy controls during NREM sleep ($168 \pm 9$ ms—Fig. 1d). On the contrary, in healthy awake controls, PLF persisted until $248 \pm 12$ ms when stimulating parietal cortex and $248 \pm 15$ ms when stimulating frontal cortex (Fig. 1d and Supplementary Fig. 2C).

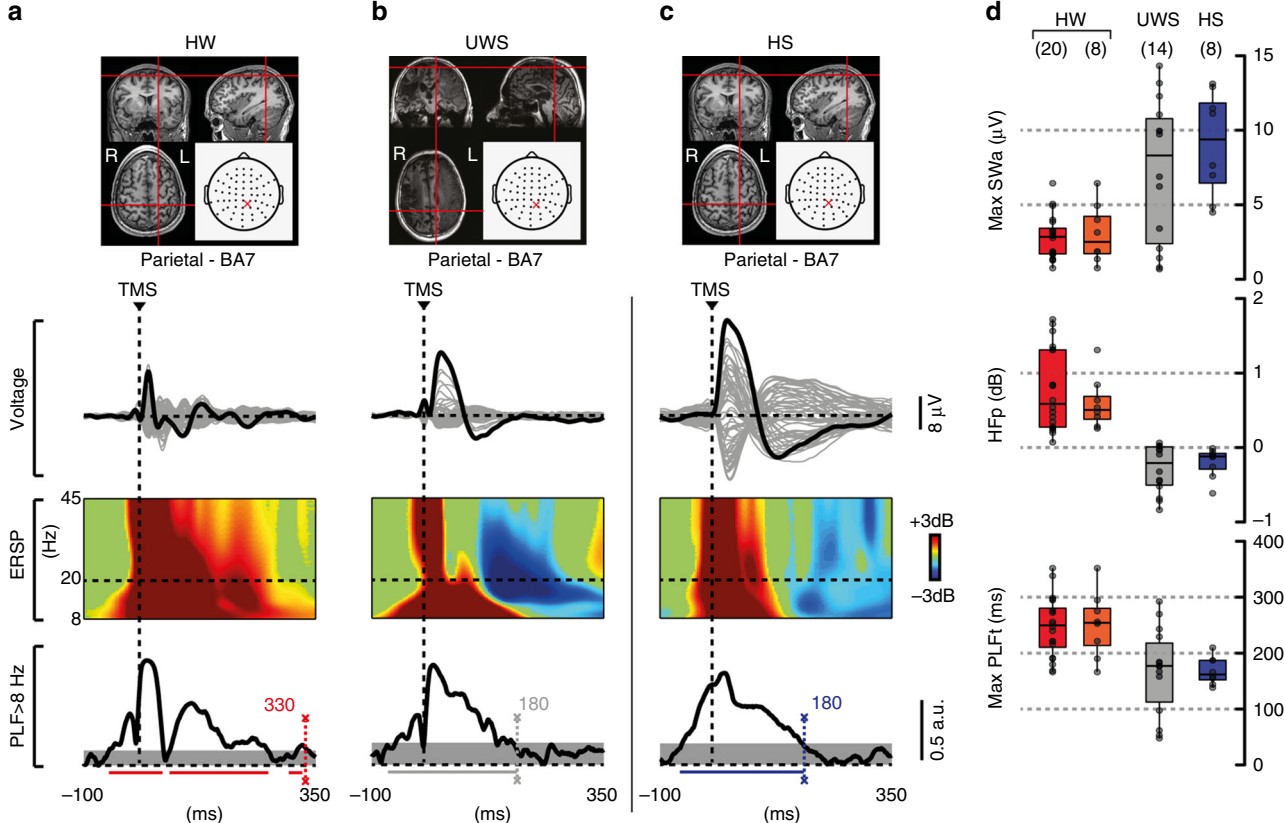

**Fig. 1** TMS evokes a sleep-like OFF-period and an early drop of PLF in UWS patients. Results for a representative healthy subject during wakefulness (HW) and NREM sleep (HS) and a representative UWS patient (patient 11 in Supplementary Table 2) are shown for parietal stimulations (BA7). **a–c** MRIs and cortical targets as estimated by the Navigated Brain Stimulation system are shown (top). A dashed vertical line marks the occurrence of TMS. Butterfly plots of the TMS-evoked EEG potentials recorded at all 60 channels (gray traces) are depicted. Event-related spectral perturbation (ERSP) and PLF are presented for the electrode with the larger response (black trace). In the ERSP plot, significance for bootstrap statistics is set at $\alpha < 0.05$ (absence of any significant activation is colored in green): statistically significant increases of power compared to baseline are colored in red, while blue represents significant power decreases. The dashed horizontal line indicates the 20 Hz frequency bin. PLF time points above statistical threshold (gray shaded area) are indicated at the bottom by a colored horizontal line. The colored-dashed vertical line indicates the timing of the last significant ($\alpha < 0.01$) PLF time point. **d** From top to bottom, boxplots of slow wave amplitude (max SWa), high-frequency power (HFp), and duration of PLF (max PLFt) for HW (red and orange), HS (blue) and UWS (gray) are shown. Boxplot displays the median (center line), the first and third quartiles (bounds of box). The whiskers extend from the bound of the box to the largest/smallest value no further than 1.5* inter-quartile range. Outlier datapoints are indicated by dots outside whiskers

These results were statistically significant at the group level, whereby max PLFt was significantly shorter in UWS patients (Wilcoxon ranksum test, $P = 0.003$ and $P = 0.031$ for parietal and frontal stimulation, respectively; Fig. 1d and Supplementary Fig. 2C, bottom panel; Table 1 and Supplementary Table 1) and healthy subjects during NREM sleep in comparison to healthy awake subjects (Wilcoxon signrank test, $P = 0.016$; Fig. 1d, bottom panel; Table 1).

Next, we asked whether the three distinctive features of the cortical response found in UWS (i.e. the presence of a slow wave-like response, high-frequency (>20 Hz) suppression and shorter PLF duration) were related. These variables are thought to reflect neurophysiological events (such as the level of neuron membrane polarization, the degree of neuronal silencing and its impact on deterministic responses) that are causally linked and showed significant correlation in previous intracranial[10] and extracranial[25] studies. In order to demonstrate this relationship, we computed linear correlations between max SWa and the maximum level of high-frequency (>20 Hz) suppression (max SHFp) and between the timing of max SHFp (max SHFt) and max PLFt, respectively. Interestingly, max SWa was significantly correlated with max SHFp ($R^2 = 0.4$, $P = 9.927*10^{-4}$; Fig. 3 left).

Also, max SHFt was significantly correlated with max PLFt ($R^2 = 0.34$, $P = 4.765*10^{-4}$; Fig. 3 right), showing that (i) larger evoked slow waves corresponded to more pronounced OFF-periods and (ii) earlier OFF-periods corresponded to an earlier dampening of the causal effects induced by the initial activation.

**OFF-periods reduce global complexity in UWS patients**. We finally asked whether cortical OFF-periods and their aftermath on local causality might be responsible for the loss of global brain complexity. All UWS patients included in the present study were characterized by levels of brain complexity (PCI range: 0.13–0.30) invariably lower than the ones measured in healthy awake subjects (PCI range: 0.32–0.64; Wilcoxon ranksum test, $P = 2.911*10^{-11}$). Notably, these lower PCI values could be explained by a difference in the time-course of the build-up of brain complexity after TMS (PCI(t), see Methods section). While in awake healthy subjects PCI(t) kept growing up to about 300 ms (272 ± 4.6 ms, mean ± SEM, Fig. 4a, right plot), in low complexity UWS patients PCI(t) grew initially but reached a plateau at an earlier time point (197 ± 12 ms) resulting in a significantly shorter build-up (Wilcoxon ranksum test, $P = 1.400*10^{-6}$). Most relevant, the timing at which global complexity stopped growing

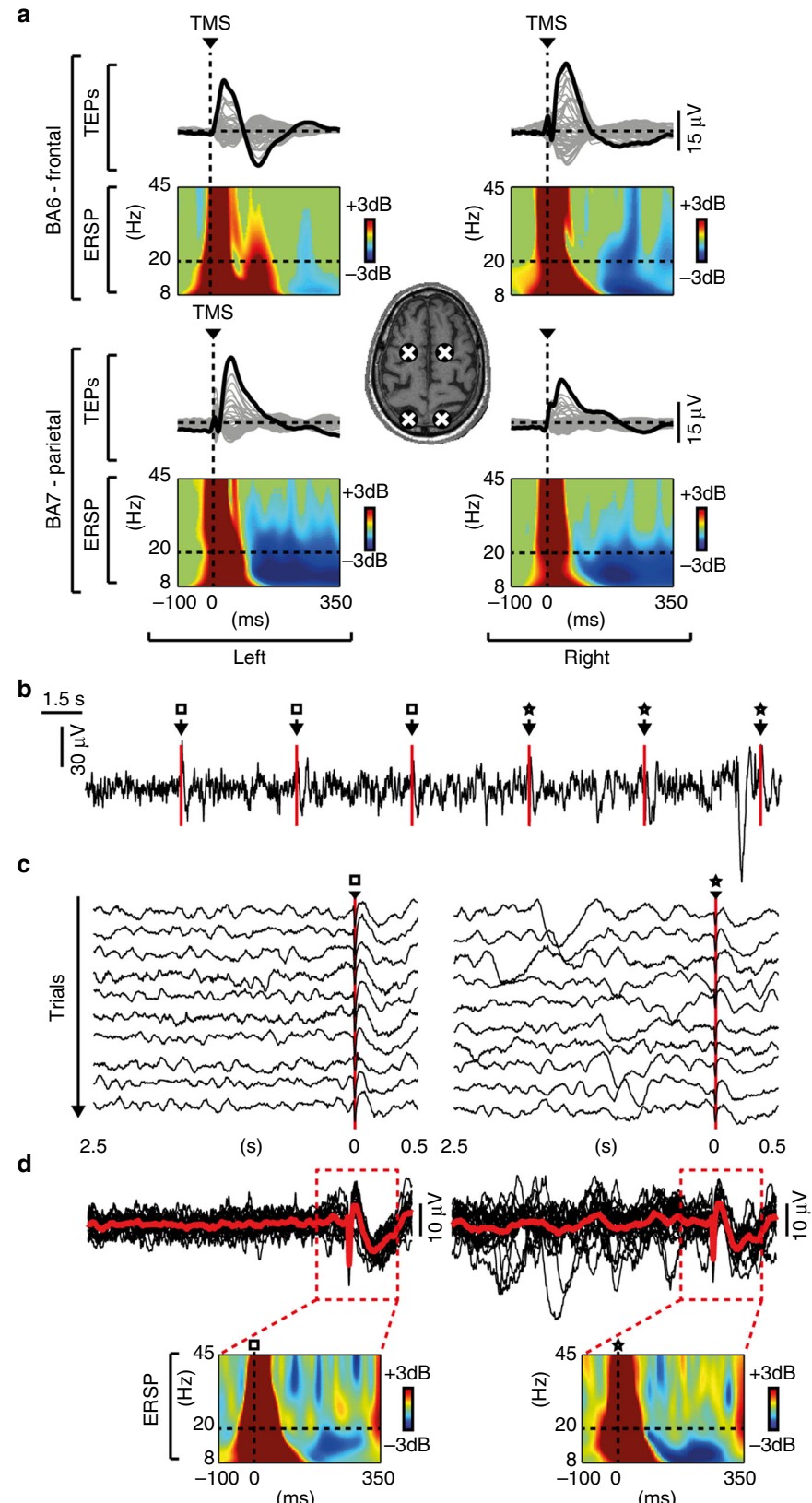

(max PCIt) showed a significant positive correlation with the timing of the OFF-period (max SHFt; $R^2 = 0.46$, $P = 3.034 \times 10^{-4}$; Fig. 4c upper plot) as well as with the timing at which local causality broke-off (max PLFt; $R^2 = 0.56$, $P = 2.974 \times 10^{-5}$; Fig. 4c lower plot). This result is highlighted in Fig. 4b for a representative UWS patient (Patient 12), where the time courses of high-frequency EEG power modulation, broadband PLF and PCI are depicted. To further strengthen the link between OFF-periods, loss of local causality and global complexity, we observed that recovery of consciousness (as assessed by the CRS-R) in a

**Fig. 2** TMS evokes an OFF-period at all targeted sites and irrespective of pre-stimulus activity. **a** White crosses on the structural MRI indicate the cortical TMS targets (BA6-frontal/BA7-parietal and left/right) in patient 15. For each cortical target, butterfly plots of the TMS-evoked EEG potentials recorded from all 60 channels (gray traces) are shown. The electrode with the largest TMS-evoked EEG potential is highlighted (black trace) and the corresponding ERSP is presented. The dashed horizontal line marks the 20 Hz frequency bin and the dashed vertical line indicates the occurrence of TMS. **b** EEG activity (one representative electrode -Cz- re-referenced to the mathematically linked mastoids) recorded in patient 4 while TMS was delivered with an inter-stimulus interval randomly jittering between 5000 and 5300 ms. Empty squares and stars indicate TMS pulses delivered over an ongoing activity showing (stars) or not showing (empty squares) spontaneous slow waves. **c** Similar to **b**, the empty square and the star indicate trials in which TMS pulses were delivered over an ongoing activity, respectively, showing or not showing spontaneous slow waves. **d** The same trials shown in **c** are superimposed and averaged (red lines) for both conditions. The corresponding ERSPs are shown in the bottom panels

| Table 1 Statistical analyses performed between groups stimulated over BA7 | | | |
|---|---|---|---|
| | Wilcoxon ranksum test (P) | Wilcoxon ranksum test (P) | Wilcoxon signrank test (P) |
| | HW ($N = 20$) vs. UWS ($N = 14$) | UWS ($N = 14$) vs. HS ($N = 8$) | HW ($N = 8$) vs. HS ($N = 8$) |
| max SWa | 0.014 | 0.357 | 0.008 |
| HFp | $1.053*10^{-6}$ | 0.999 | 0.008 |
| max PLFt | 0.003 | 0.785 | 0.016 |

Statistical comparison regarding boxplots of slow wave amplitude (max SWa), High-Frequency power (HFp), and duration of PLF (max PLFt) presented in Fig. 1d. Details regarding the applied tests, the sample size and the significance values for each comparison between conditions (HW, UWS, HS) are reported

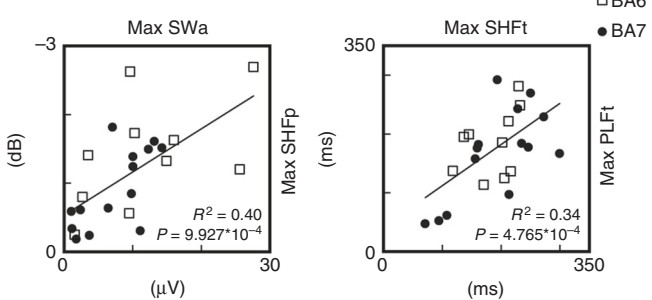

**Fig. 3** Slow-wave amplitude, high-frequency suppression and PLF duration correlate in UWS patients. On the left, the correlation between the maximal amplitude of the evoked slow wave (max SWa) and the maximal level of high-frequency (>20 Hz) power suppression (max SHFp) is shown. On the right, the correlation between the timing of the maximum high-frequency suppression (max SHFt) and the timing of the last significant time point of phase-locking (max PLFt) is shown. For both correlations, the coefficient of determination $R^2$ and the significance level $P$ are reported. White squares and black dots represent BA6 and BA7 stimulation, respectively

longitudinally recorded patient (Patient 16) was paralleled by a progressive reduction of high-frequency (>20 Hz) suppression, a concurrent prolongation of PLF and an increase of PCI up to values found in conscious awake subjects (Fig. 5).

## Discussion

Previous studies employing TMS/EEG have shown that most UWS patients retain portions of the cerebral cortex that are active and reactive[26–28] but blocked in a state of low complexity[7,9]. Here we investigated the electrophysiological mechanisms underlying this condition; we show that in these low-complexity patients the cortical response to TMS is underpinned by an OFF-period. Further, we demonstrate that the occurrence of this OFF-period rapidly disrupts the build-up of causal effects of the initial activation thus preventing the emergence of large-scale complex interactions. Similar electrophysiological events were detected in sleeping healthy controls but were never found in healthy awake subjects, suggesting that a pathological form of sleep-like OFF-periods may occur in UWS patients. Therefore, the present findings link cortical bistability—a phenomenon with a well-characterized neuronal mechanism that is known to play a role in physiological NREM sleep—to the pathophysiology of the UWS.

At the cortical level, the key feature of NREM sleep is the occurrence of OFF-periods, reflecting a profound hyperpolarization in the membrane of cortical neurons. This phenomenon, often referred to as cortical bistability, is caused by the enhancement of adaptation (or activity-dependent) $K^+$ currents, brought about by decreased levels of neuromodulation from brainstem activating systems[29–32] and/or by increased inhibition[16,17]. Due to these mechanisms, cortical neurons tend to plunge into a silent, hyperpolarized state, lasting few hundreds of milliseconds, after an initial activation[13,33]. In the sleeping brain, the occurrence of synchronous membrane hyperpolarization in cortical neurons is reflected at the extracellular level in large slow waves associated with transient suppressions of high-frequency (>20 Hz) activity that may be detectable in spontaneous activity

both in the local field potential[22,23] and in the EEG[24,25]. However, due to its activity-dependent nature, bistability and the associated OFF-periods can be better revealed using a perturbational approach, whereby the impulse-response properties of cortical neurons is probed by means of direct activations. Hence, intra-cortical stimulations have been employed to investigate cortical bistability in humans and its effect on the propagation of cortico-cortical evoked potentials during wakefulness and sleep[10,11].

A key finding of the present work is the demonstration of a pathological form of sleep-like OFF-periods in the brain of UWS patients (Fig. 1b and Supplementary Fig. 2B). Specifically, targeting neuronavigated TMS to intact portions of both their frontal and parietal cortices invariably elicited a stereotypical slow wave associated with a high-frequency (>20 Hz) suppression activity matching that of healthy sleeping subjects (Fig. 1d and Supplementary Fig. 2C). Notably, these sleep-like OFF-periods were never found when the same cortical areas were stimulated in awake healthy subjects (Fig. 1a and Supplementary Fig. 2A).

Why do awake brain-injured patients show cortical responses that are typical of the sleeping brain? A possibility is that structural lesions may lead to functional changes that enhance adaptation mechanisms and hence the tendency of intact portions of the thalamocortical system to transiently fall into a quiescent OFF-period[34,35]. For example, this might happen when sub-cortical lesions, such as diffuse axonal injury, interrupt a critical amount of fibers of the ascending activating systems[36]. In an extreme case, the thalamocortical system may be largely intact but functionally constrained to a pathological tendency towards OFF-periods due a predominance of adaptation currents[31,37].

Multifocal white matter lesions and diffuse axonal injury may also induce bistability by engendering a state of cortico-cortical

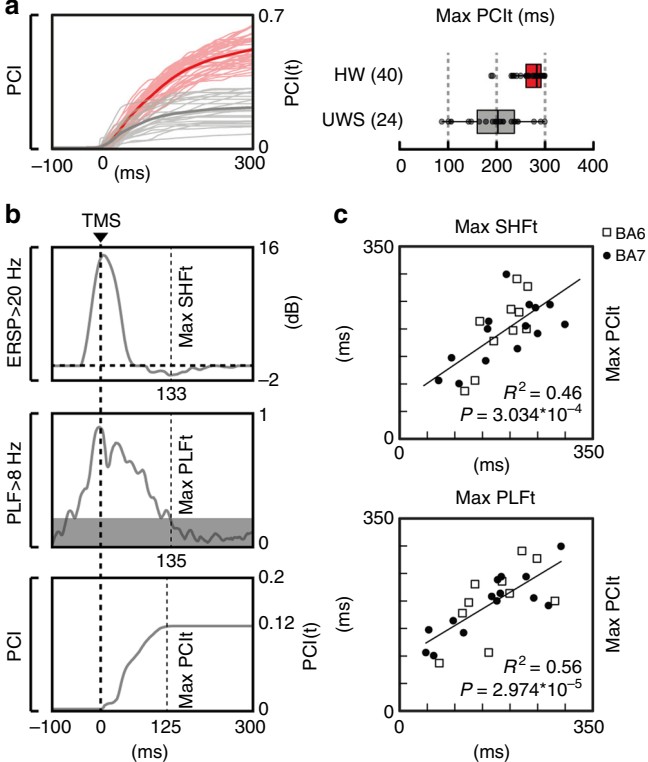

**Fig. 4** The occurrence of an OFF-period prevents the build-up of PCI. **a** For each TMS/EEG measurement, the temporal evolution of PCI, i.e. PCI(t), calculated in HW (thin red lines) and UWS patients (thin gray lines) are shown together with their grand average (thick lines). The boxplot shows the time at which PCI reaches its maximum value (max PCIt) for HW and UWS patients. The boxplot displays the median (center line), the first and third quartiles (bounds of box). The whiskers extend from the bound of the box to the largest/smallest value no further than 1.5* inter-quartile range. Outlier datapoints are indicated by dots outside whiskers. **b** The time course of the high-frequency power averaged above 20 Hz (top) and the significant PLF above 8 Hz (middle) of the electrode with the largest response, together with the temporal evolution of PCI (bottom) are shown for one representative patient (patient 12 in Supplementary Table 2). Thin dashed vertical lines mark the timing of the maximum high-frequency suppression (max SHFt, top), the timing of the last significant time point of phase-locking (max PLFt, middle) and the time at which PCI reach its maximum value (max PCIt, bottom), respectively. Thick dashed vertical line indicates the occurrence of TMS. **c** The correlation between max SHFt and max PCIt (top) and the correlation between max PLFt and max PCIt (bottom) are shown. The coefficient of determination $R^2$ and the significance $P$ level are reported. White squares and black dots represent values corresponding to BA6 and BA7 stimulation, respectively

disfacilitation, that is by reducing recurrent excitation[20]. Indeed, intracellular recordings have shown that, following a surgical white matter undercut (cortical slab), pyramidal neurons can switch their discharge patterns from tonic firing to an intrinsically bursting regime promoting the alternation between periods of intense firing and silence[35,38–40]. Crucially, a critical reduction of cortico-cortical connectivity may shift the excitation/inhibition balance, leading to cortical OFF-periods by excessive inhibition[16,19]. This is known to occur locally after a stroke[41], but may involve the whole remaining cortex following severe, multifocal injury[42].

Finally, multifocal brain injury can also induce critical functional shifts by altering the balance within the cortico-striatal mesocircuits[43,44]. This mechanism is particularly relevant because it may lead to both cortical disfacilitation and thalamic hyperpolarization. Importantly, if the latter exceeds a given threshold, thalamic neurons may switch their firing pattern from tonic to bursting mode[45], thus further enhancing cortical bistability[46].

Overall, different mechanisms, alone or in combination, may engender a tendency towards cortical bistability in brain-injured patients, as also reflected by the presence of slow waves in their spontaneous waking EEG (Supplementary Fig. 3)[47–49]. While their relative contribution is difficult to disentangle, it is worth noting that all the above mechanisms can be effectively engaged by a cortical perturbation. For example, a direct cortical hit with TMS may (i) trigger activity-dependent $K^+$ currents and an OFF-period, if $K^+$ channels are de-inactivated; (ii) massively recruit local inhibitory circuits leading to an OFF-period, if the excitation–inhibition balance is biased towards the latter; (iii) force hyperpolarized thalamocortical neurons to fire bursts of action potentials back to the cortex and then fall into a prolonged silence, when these cells are in a bursting mode. In fact, TMS perturbations could reveal the presence of adaptation mechanisms and of the ensuing OFF-periods in all patients, regardless of their background EEG pattern (Supplementary Table 2), of the prevalence of spontaneously occurring slow waves (Supplementary Fig. 3), and of pre-stimulus ongoing activity (Fig. 2b–d).

Intriguingly, the strength of adaptation mechanisms has been considered as a key factor in shaping the behavior of dynamical systems that are employed to model sleep-like activity[15,50,51]. In future studies, it would be crucial to explore the formal relationships between cortical bistability as experimentally observed here and the notion of bistability as a system-wide phenomenon defined in the framework of dynamical systems[52,53].

Besides the specific mechanisms engendering cortical OFF-periods, it is relevant to consider their large scale consequences in brain-injured patients. In UWS patients OFF-periods were ubiquitously observed for TMS applied over parietal and frontal cortices (Figs. 1 and 2 and Supplementary Fig. 2). In this way, bistability obliterated the physiological differentiation of the impulse response across cortical areas (i.e. the natural frequency[54]) that is normally observed in awake, conscious subjects. At the same time, OFF-periods curtailed the duration of deterministic responses, as revealed by the correlation between the timing of their occurrence (SHFt) and the abrupt termination of phase-locked oscillations (PLFt). This finding is in line with the results of in vitro[14] and in vivo[20] observations suggesting that the resumption of cortical activity following an OFF-period is a stochastic process. To the extent that recurrent interactions rely on the amplification of coherent activity across distributed sets of neurons, the scrambling of phases operated by the OFF-periods at each node may critically impair the emergence of large-scale cortical integration[55].

In view of the above, we assessed the relationships between the occurrence of OFF-periods, the duration of phase-locking and the temporal evolution of PCI, an index that is explicitly designed to quantify the joint presence of differentiation and integration in cortical networks[9], a crucial requirement for consciousness according to theoretical neuroscience[56]. In UWS patients the build-up of complexity (max PCIt) was shorter and never reached the levels attained in awake, conscious subjects; crucially, the time at which PCI stopped growing correlated significantly with both the occurrence of the OFF-period (maxSHFt) and the termination of phase-locked activity (PLFt) (Fig. 3). These results corroborate the hypothesis that bistability and OFF periods may be in a key position to impair overall brain complexity. Most important, these significant correlations draw a first link between neuronal events and global brain dynamics relevant for pathological loss and recovery of consciousness.

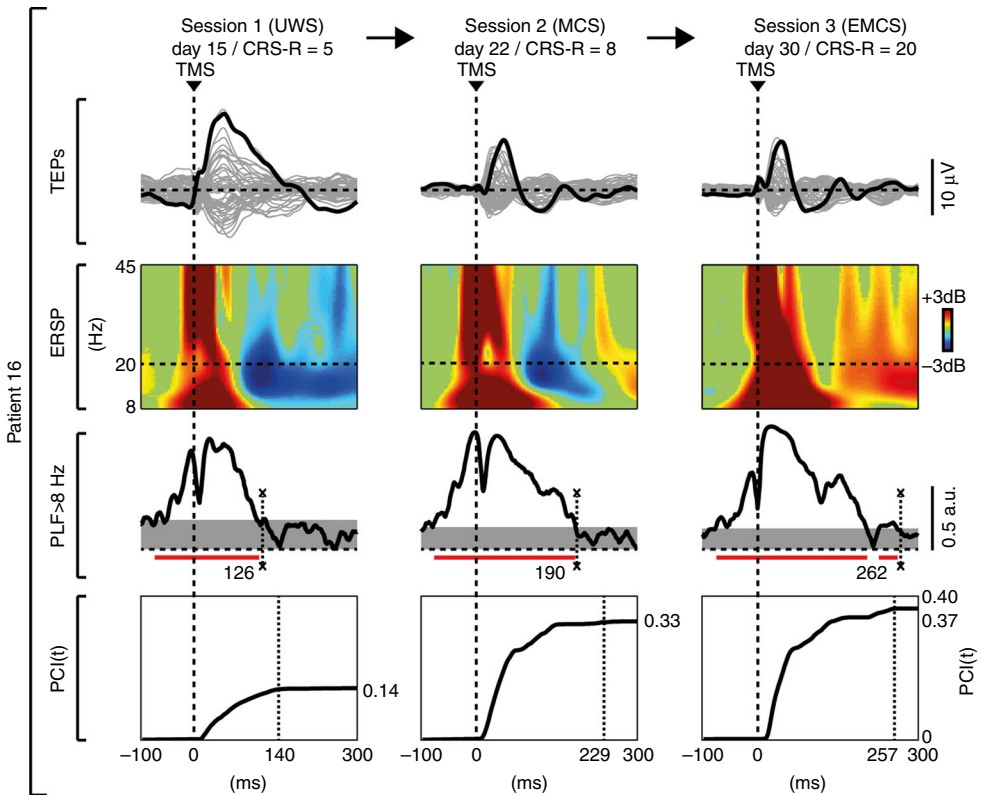

**Fig. 5** Longitudinal measurements in one UWS patient who evolved to EMCS, through MCS. In Patient 16 (see Supplementary Table 2) the first behavioral and TMS/EEG assessments (Session 1) were carried out 48 h after withdrawal of sedation, as patient exited from coma[26]. The butterfly plot of the TMS-evoked EEG potentials recorded from all 60 channels (gray traces), the corresponding ERSP and the PLF time course of the channel with the largest response are shown for each clinical diagnosis (UWS, MCS, and EMCS) together with the temporal evolution of PCI. In the ERSP plot, red color indicates a significant ($\alpha < 0.05$) power increase compared to the baseline, blue color a significant power decrease and the green color a non-significant activation. The dashed horizontal line marks the 20 Hz frequency bin. The last significant ($\alpha < 0.01$) time point in the PLF (above 8 Hz) is marked by a thin dashed vertical line. Time points above statistical threshold (gray shaded area) are underlined by a red horizontal line. The thick dashed vertical line indicates the occurrence of TMS. The time at which PCI reaches its maximum value (max PCIt) is indicated by a thin dashed vertical line

Previous studies have shown that loss of consciousness in UWS patients is associated with a variable degree of brain damage and physical disconnection of neural linkages[57,58]. In a minority of brain-injured patients plastic structural changes, including axonal regrowth, may directly support behavioral recovery[59,60]; in others cases, functional adjustments may play a major role, while the amount of structural brain damage remains substantially equal[61–63]. In this respect, to the extent that pathological sleep-like bistability represents a common functional endpoint disrupting large-scale interactions across structurally intact portions of the cortex, its reversal may potentially be relevant for clinical recovery.

The course of events illustrated in Fig. 5 is compatible with this hypothesis. This figure illustrates the results of longitudinal TMS/EEG measurements performed in one patient evolving from UWS to minimally conscious state (MCS), and eventually regaining consciousness. In this patient, behavioral recovery occurred in the space of two weeks and was associated with a progressive decrease of bistability and a concurrent recovery of causality and complexity.

As a proof of principle, a recent microscale study employing electrical stimulation and recordings in isolated cortical slices showed that phase-locking and complex causal interactions, as assessed by an adapted version of PCI, could be effectively restored by pharmacological interventions that reduce bistability and increase cortico-cortical excitability[64]. This microscale finding further suggests a causal link between cortical bistability and complexity and may have translational implications since brain slices can be considered a simplified model of the electrophysiological state of the cerebral cortex under conditions of severe deafferentation.

While elucidating the mechanisms of recovery is clearly beyond the scope of this work, the present observations in UWS patients raise important questions. Can neuromodulation or pharmacological manipulation push neurons beyond the threshold for bistable dynamics thus promoting recovery of complexity? Are some patients just below this critical threshold? Different interventions, such as zolpidem or amantadine administration[65,66], thalamic stimulation with deep brain stimulation[62] or low-intensity focused ultrasound pulsation[63], transcranial direct current stimulation[67], and vagal nerve stimulation[68] have demonstrated significant behavioral improvements in individual patients but a reliable read-out and interpretation of their end-point effects at the level of cortical circuits is still lacking. In the view of the present results, detecting the presence of cortical sleep-like bistability and tracking its evolution over time, may offer an objective reference to devise, guide, and titrate therapeutic strategies aimed at restoring consciousness. In this respect, it will be crucial to further elucidate the relationships between cortical bistability, neuronal OFF-periods and overall network complexity through extensive experiments across scales, species, and models, spanning from ionic channel modeling to whole-brain simulations and macroscale measurements at the patient's bedside.

## Methods

**Study design.** Here, we tested the hypothesis that sleep-like cortical OFF-periods characterizes the cortical response to TMS in UWS patients by analyzing 72 TMS/EEG measurements in 36 subjects.

According to an open-label design, we first compared at the group level TMS-evoked EEG potentials recorded in 16 ($N = 16$) awake severely brain-injured patients diagnosed with a UWS (Supplementary Table 2) with PCI < 0.31 and 20 healthy volunteers ($N = 20$) in wakefulness and NREM sleep (Supplementary Table 3). Specifically, from TMS-evoked EEG potentials we derived the EEG power <4 Hz (SWa) and >20 Hz (HFp) to detect the presence of cortical OFF-periods, and compared these two indices between awake healthy subjects, awake UWS patients, and healthy subjects during NREM sleep. Then, we calculated the duration of the broadband PLF (>8 Hz), which accounts for the impact over time of the TMS on the phase of the EEG response. Finally, we compared the time course of PCI in an open-label design between healthy awake subjects and awake UWS patients. For details about the recruitment criteria of study participants and the analysis of TMS-evoked EEG potentials see "Protocols and procedures" and "Data Analysis" sections, respectively.

**Protocols and procedures.** Patients underwent multiple behavioral assessments by means of the CRS-R; for a period of one week (four times, every other day), and one session of electrophysiological recording (TMS/EEG and spontaneous EEG at rest) within the same evaluation week. One additional patient (Patient 16 in Supplementary Table 2) was clinically monitored for a period of 1 month and underwent three neurophysiological assessments: the first while in a UWS condition, then after clinical evolution to a MCS, and eventually upon emergence from the minimally conscious state (EMCS) as assessed by the CRS-R. All UWS patients were included in the low-complexity UWS subgroup described in a recent study[7]. A single recording session was performed in each patient, except for the longitudinal assessment performed in Patient 16. During the recording, UWS patients were lying in their beds with eyes open, and vigilance was continuously monitored. In case signs of drowsiness appeared, recordings were temporarily interrupted and patients were stimulated using the CRS-R arousal facilitation protocols[21]. TMS targets were selected bilaterally within the frontal and the parietal cortices (Brodmann area—BA6 and BA7, respectively) based on the individual anatomical MRI[7] and the precision and reproducibility of stimulation were controlled using a Navigated Brain Stimulation system (Nexstim Ltd., Finland). The need to avoid direct stimulation of cortical lesions guided the specific selection of TMS targets[27] in UWS patients. Depending on the spatial extent and location of lesions in each individual patient, in the present study we considered 24 TMS/EEG measurements, from 16 UWS patients, obtained by stimulating either one cortical site (BA6 or BA7) or both (see Supplementary Table 2).

Healthy volunteers (see Supplementary Table 3) underwent a general medical and neurological examination in order to prevent potential adverse effects of TMS, and exclude major medical and/or neurological diseases as well as substance abuse. All healthy subjects were recorded during wakefulness with eyes open while lying on a reclining chair with a headrest to ensure a stable head position. In 20 healthy subjects, both BA6 and BA7 were targeted with TMS either on the left or on the right side, counterbalanced across individuals. In a subgroup of eight healthy subjects we performed TMS/EEG measurements by targeting BA6 and BA7 during wakefulness before lights off, while as soon as the participant reached a stable N3 sleep stage[69], only BA7 was stimulated (see Supplementary Table 3). This choice was dictated by the previous observation that during NREM sleep TMS evokes larger EEG responses in parietal areas as compared to frontal sites[70].

The intensity of the TMS-induced electric field was always set above 120 V/m based on the neuronavigation system. The intensity of 120 V/m has been shown to generate significant and reproducible TMS-evoked EEG potentials[71]. Overall, the TMS-induced electric field was comparable between UWS patients (124.8 ± 9.8 and 138.3 ± 12.9 V/m, mean ± SEM, for parietal and frontal stimulation, respectively) and awake healthy subjects (132.9 ± 4.8 and 133.9 ± 5.5 V/m, Wilcoxon ranksum test, $P = 0.452$ and $P = 0.813$, for parietal and frontal stimulation, respectively). In healthy subjects who underwent TMS/EEG measurements both during wakefulness and sleep, the same stimulation parameters were applied by means of the Navigated Brain Stimulation system. For all the TMS/EEG measurements, the location of the maximum electric field induced by TMS on the cortical surface (hotspot) was always kept on the convexity of the targeted cortical gyrus with the induced current perpendicular to its main axis. In each TMS/EEG measurement, at least 200 stimulation pulses were delivered with an inter-stimulus interval randomly jittering between 2000 and 2300 ms (0.4–0.5 Hz). For a detailed description of the TMS and EEG equipment see Supplementary Methods.

All the experimental procedures were approved by the following ethical committees: Istituto di Ricovero e Cura a Carattere Scientifico Fondazione Don Gnocchi Onlus, Milan, Italy; Comitato Etico Milano Area 1, Milan, Italy; Comitato Etico Milano Area 3, Milan, Italy; Medicine Faculty of the University of Liège, Liège, Belgium. All healthy participants gave written informed consent, while for non-communicating UWS patients the informed consent was obtained by a legal surrogate.

**Spontaneous EEG classification in UWS patients.** Rest EEG recordings collected in UWS patients were evaluated according to a clinical classification recently proposed[72] after bandpass filtering between 1 and 70 Hz, downsampling to 725 Hz, and re-referencing to the standard longitudinal montage. The EEG category of each patient is reported in Supplementary Table 2.

**Data analysis.** Data analysis was performed using Matlab R2012a (The Math-Works Inc.). TMS/EEG recordings were visually inspected to reject trials and channels containing noise or muscle activity[7,27]. Then, EEG data were bandpass filtered (1–45 Hz, Butterworth, third order), down-sampled to 725 Hz and segmented in a time window of ±600 ms around the stimulus. Bad channels were interpolated using the spherical function of EEGLAB[73]. Recording sessions with either more than 10 bad channels or less than 100 artifact-free trials were excluded from further analysis. Then, trials were re-referenced to the average reference and baseline corrected. Finally, independent component analysis (ICA) was applied in order to remove residual eye blinks/movements, TMS-evoked and spontaneous scalp muscle activations.

In order to characterize TMS-evoked EEG potentials we (1) measured the amplitude of a TMS-evoked slow wave (<4 Hz) and then (2) detected the occurrence of a cortical OFF-period by quantifying the amount of significant suppression of high-frequency (>20 Hz) EEG power compared to pre-stimulus[10,22–24]. Operationally, for each EEG channel $i$ (1–60), we followed the stepwise procedure presented in Supplementary Fig. 1 and described below:

(1) To assess the amplitude of TMS-evoked slow waves, single trials were low-pass filtered below 4 Hz (third-order Chebyshev filtering as in ref. [10]), re-referenced to the mathematically linked mastoids, averaged and eventually rectified. For each channel $i$, the maximum Slow Wave amplitude (max SWa(i)) was computed as the maximum amplitude of the rectified signal within the 8–350 ms time window (Supplementary Fig. 1A).

(2) To assess the suppression of high-frequency (>20 Hz) EEG power, we applied the event-related spectral perturbation (ERSP) routine implemented in EEGLAB[73]. Specifically, single trials were time–frequency decomposed between 8 and 45 Hz using Wavelet transform (Morlet, 3.5 cycles; as in ref. [54]) and then normalized with the full-epoch length (here ranging from −350 to 350 ms) single-trial correction[74]. The resulting ERSPs were averaged across trials and baseline corrected (from −350 to −100 ms). Furthermore, power values that were not significantly different from the baseline were set to zero. To detect statistically significant activation in the time–frequency domain we applied a bootstrap statistics ($\alpha < 0.05$), with a number of permutations = 500. Finally, the time course of the significant high-frequency EEG power was obtained by averaging over frequency the ERSP values above 20 Hz[22]. Then, from the time course of significant high-frequency EEG power of each channel $i$, we extracted three parameters: the integral between 100 and 350 ms of the high-frequency (>20 Hz) power (HFp(i)), the maximum value of high-frequency power suppression (max SHFp(i)) and the timing of the maximum high frequency (>20 Hz) power suppression (max SHFt(i)).

All the measures described above (max SWa(i), HFp(i), max SHFp(i) max SHFt(i)) and calculated at the single channel level were averaged over the four channels closer to the stimulation site (Supplementary Fig. 1D) and the resulting averages were labeled: max slow wave amplitude (SWa), high-frequency power (HFp), max SHFp (maximum value of Suppression of High-Frequency power) and max SHFt (timing of the maximum value of Suppression of High Frequency).

The impact of the OFF-periods on local causal interactions was assessed by means of broadband (>8 Hz) PLF[75]. PLF can be calculated for every single electrode as an adimensional index (range 0–1) defined as the absolute value of the average of the Hilbert Transform across trials. To the extent that instantaneous PLF (i.e. the time-course of PLF) measures the coherence of the response to a perturbation across trials in a specific time-window, it can be used to quantify the duration of the deterministic effect of a given input[10,75]. Here, for each EEG channel $i$ (1–60), single trials were high-pass filtered above 8 Hz (third-order Butterworth filter) and PLF was computed as the absolute value of the average of the Hilbert Transform of all single trials. Assuming a Rayleigh distribution of the baseline values from −500 to −100 ms, PLF time points that were not significantly different from baseline ($\alpha < 0.01$) were set to zero. For each channel $i$, the latest significant PLF time point was identified and labeled as max PLFt(i). Finally, max PLFt (timing of the last significant time point of phase-locking) was calculated as the average of max PLFt(i) over the four channels closer to the stimulation site (Supplementary Fig. 1D).

Finally, in order to assess the effects of bistable dynamics on the complexity of global causal interactions, we first half-sampled the data and then we calculated PCI by applying a fully automatic procedure[7,9]. Specifically, after source modeling (three spheres BERG method as conductive head volume, weighted minimum norm constraint applied to an "empirical" Bayesian approach), non-parametric bootstrap-based statistical analysis was performed to extract the significant spatiotemporal pattern of the TMS-evoked responses. Then, PCI was obtained as the Lempel–Ziv complexity of the matrix of significant cortical source activity and normalized by source entropy, resulting in a positive real number between 0 and 1 (minimally and maximally complex patterns, respectively). To further study the relationships between bistable dynamics and the emergence of complex interactions, we used the temporal evolution of PCI, i.e PCI(t), which describes the buildup of complexity of the deterministic brain responses to TMS over time.

Specifically, we rounded PCI(t) to the second decimal place and we measured the first time point in which PCI(t) reached its maximum (max PCIt).

**Statistical analysis**. Group analyses were performed in Matlab R2012a by using Wilcoxon ranksum test and Wilcoxon signrank test where appropriate (see Table 1 and Supplementary Table 1 for details).

## Data availability

The data that support the findings of this study are available from the authors on reasonable request; see author contributions for specific data sets.

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

## Acknowledgements

This research has received funding from the European Union's Horizon 2020 Framework Program for Research and Innovation under the Specific Grant Agreement No. 720270 (Human Brain Project SGA1) (to M.M., M.R., and S.L.) and No. 785907 (Human Brain Project SGA2) (to M.M., M.R., and S.L.) and by the EU grant H2020, FETOPEN 2014-2015-RIA no. 686764 "Luminous" (to M.M. and S.L.). The study has been also funded by the grant "Sinergia" CRSII3_160803/1 of the Swiss National Science Foundation (to M.M.), by the James S. McDonnell Foundation Scholar Award 2013 (to M.M.), and by the Tiny Blue Dot Foundation (to M.M.). Finally, this study has been partially funded by the grant "Giovani Ricercatori" GR-2011-02352031 of the Italian Ministry of Health (to M.R. and S.S.), by the Belgian Funds for Scientific Research (FRS) (to S.L.), by the "Fondazione Europea di Ricerca Biomedica" (to S.L.), and by the grant #2016/08263-9 of the São Paulo Research Foundation (FAPESP) (to A.G.C.). We are grateful to Chiara Cirelli and Giulio Tononi for their comments and suggestions. We thank Anna Cattani, Thierry Nieus, Maurizio Mattia, Ezequiel Mikulan, and Giovanni Casazza for their help and comments. We also thank Simone Russo, Sasha D'ambrosio, Chiara-Camilla Derchi, and Alice Mazza for critically reading the manuscript.

## Author contributions

Conceptualization: M.R., M.F. and M.M.; designed and performed experiments in patients: M.R., M.F., S.C., A.C., O.G., O.B. and S.L.; clinical evaluation of patients: G.D., A.C., G.C., M.B. and O.G.; designed and performed experiments in healthy subjects during wakefulness and sleep: M.F., A.P., S.S., A.C. and M.R.; data analysis: M.F., A.G.C., A.P., F.S. and S.C.; wrote the manuscript: M.R,, M.F., S.S. and M.M.

## Additional information

**Competing interests:** The authors declare no competing interests.

