## [Peer Review File · Nature Communications]

Reviewers' comments:

Reviewer #1 (Remarks to the Author):

The current study focuses on unresponsive wakefulness syndrome (UWS); a condition where brain injured individuals do not show behavioral signs of consciousness, which the authors believe is due to an abnormal OFF-period response to transient cortical activation. Using transcranial magnetic stimulation in patients with UWS, the chief result is the cortical evoked response consists of a slow wave <4Hz and reduced power >20Hz, which resembles the EEG OFF-period response during NREM sleep, but not the response during wakefulness, in healthy individuals. In addition, EEG signal processing shows two measures, one of local causality and the other global complexity, decrease and plateau respectively during the evoked OFF period in patients with UWS. The study concludes sleep-like OFF periods could explain loss of consciousness in UWS.

This is an interesting study and presents new results of sleep-like EEG slow waves in patients with UWS. Strengths of the manuscript include good rationale, clever design using transcranial magnetic stimulation, clear presentation of results in text and figures, and interesting discussion that proposes two reasonable explanations for sleep-like OFF period in UWS. In spite of these strengths, it is curious why spontaneous slow wave were not included in the analysis, which could also indicate pathological bi-stability; the method for computing phase-locking, and how this is a measure of causal (neuronal) interactions, is unclear; and the neuronal correlates of EEG slow waves in UWS are yet to be identified, although the observations here present several hypotheses. These comments and other are discussed below.

Fig. 2 indicates spontaneous slow waves occur in patients UWS, similar to how slow waves occur during natural NREM sleep. Are the spontaneous slow waves associated with reduced high frequency power in UWS as in NREM sleep, and if it does, then could this correspond with inappropriate sleep-like OFF periods in UWS?

The methods for computing the phase-locking factor and how this measures local causal interactions is unclear. The text indicates an average Hilbert transform of the 8Hz highpass filtered evoked response was calculated, but no indication between which pairs of electrodes phase-locking was calculated or the spatial distribution of phase-locking values. Was it restricted only to the four adjacent recording sites with largest amplitude evoked response?

In regard to the perturbational complexity index (PCI), have there been any studies on the neuronal basis of PCI? Other than the behavioral correlates of PCI it would be helpful to understand the neurophysiology that correlate with PCI values.

Changes in PLF values during the evoke response illustrated in the figure appear more variable than what is described in the text. While it is clear the latest significant PLF value occur later during the healthy awake condition than UWS or NREM sleep, what about the first and other non-significant PLF time point before the latest time point?

In Fig. 4B, why is the ERSP >20Hz increasing before, and maximal at the onset of TMS? Also, why do the PLF values fall to zero at max PLFt, and PCI value plateau at 0.12 at max PCI? Is due to the methodology or reflect physiological brain activity? This should be explained or at least noted in the legend.

The manuscript includes discussion of "...neuronal OFF-periods..." and "...neuronal events..." in explaining the some of the current findings, but it is not known what is happening at the level of neurons, unlike some unit studies of NREM slow waves in presurgical patients (e.g. Nir et al. 2011).

Other comments:

Unlike EEG, MRI, NREM sleep, and maybe TMS, extensive use of uncommon abbreviations interrupts the flow of sentences and hinders readability, e.g. TMP, TEP, PCI, SPES, PLF, HFp, max SWa, max PLFt, max SHFt, max SHFp, PCIt, and CRS-R.

In Table S1, what does severe, moderate, and mild mean in EEG category?

Should spell out ERSP in first appearance in the manuscript.

Reviewer #2 (Remarks to the Author):

Authors here present an interesting study about the effect of perturbation, performed by TMS, on the brain responses of patients with unresponsive wakefulness syndrome (UWS). They demonstrate that these patients, when awake, are characterized by states that are very close to those physiologically observed in healthy participants during non-REM sleep. Thus, they suggest that this pathological sleep-like bistability with loss of causality and complexity, in the brain of UWS patients (also in cortical regions apparently spared from injuries), could be related with their difficulty in recovering full consciousness. Turning this pathological condition toward higher levels of cortical excitability could be useful for recovering of these patients.

The manuscript is interesting, and could give some new and useful information in the field. However, in my opinion, it could be improved in some sections. Thus, I have some major and minor concerns that should be managed by Authors.

Major concerns

Abstract: The present version of the abstract could be improved by using a more "structured" model. Now, it seems more like a piece of an "introduction" section of a manuscript. Please give some adjunctive information (especially about groups, methods and results), obviously compatibly with editorial guidelines of the Journal (e.g. word limit, "style" of abstracts, and so on...).

Results/Methods

It is clear how you decided to stimulate premotor or parietal cortex but it is not clear how and when you decided to stimulate left and/or right hemisphere; moreover, it is not clear in which patients you stimulated left and/or right; moreover, when considering also healthy participants this could result in additional TMS/EEG measurements (more than 72?), but, again, it is not clear when you stimulated left/right premotor/parietal cortex, in healthy subjects, during awake and/or during sleep (in the same day? Really?). Please clarify (further tables in the supplementary material could help?). Furthermore, please give further detail about how you decided to manage data: i.e., in healthy participants, data obtained from left/right premotor/parietal cortex (in the different conditions) were always fully comparable with respect to the same measurements (perhaps partial) obtained from patients? Moreover, it is not always clear if data obtained from the different cortices were, at the end, merged for analyses or not. Please clarify.

I am not always convinced that using Bonferroni correction is really informative, it depends from the context. In this case, I think that it could be not used because this study is more close to a "proof of evidence", that will deserve further confirmation. Moreover, you give not information about the "corrected" statistical thresholds. Thus, I prefer that you indicate the "real" p-values (and not simply $p < 0.05$ or $p < 0.01$): this could be more informative about the real (and evident) effects. Similarly, information about tests used (i.e. when non-parametric ANOVA was used or when Wilcoxon/Mann-Whitney was used) should be given in a more systematic way when presenting results (or in the methods section).

Methods

Please insert a reference when you speak about the hallmarks of cortical OFF-periods, and when you present the calculation of the duration of the broadband PLF. Here, you say that only eight healthy participants were measured during NREM sleep: please give further information (age, sex, etc.). Which could be the statistical implications of using a "sub-group" of healthy participants when comparing data with UWS and awake healthy subjects?

From the last paragraph at p. 16, it seems that, even if the same lobe was stimulated, the cortical targets may be, at the end, slightly different in UWS and healthy participants. Please clarify implications of this.

Please give further information about how you decided to set intensity of stimulation.

Why did you use a disproportionate number of males/females in the two groups? Could this have an effect in terms of data "noise"? Similarly, it could be useful to have some additional data about the age of every healthy participant.

Correlations were performed (separately) in every group or by "merging" data of every group?

Discussion

Discussion is well written and structured, but it seems that (sometimes) speculations are very "invasive" (e.g. toward the end of p. 11 or in the first four/five rows of p. 12) Please try to avoid this.

Minor concerns:

Introduction section:

The introduction is informative but concise. However, please better define in this section the concept of low-complexity UWS patients (p. 4), also for less expert readers.

Results:

Just a curiosity: it could be useful/possible to correlate also SWa/maxSHFp with maxSHFt/PLFt?

Methods

Here you write that interstimulus interval was 2000-2300 ms. In figures you indicated 5000-5300 ms. Please clarify.

Supplementary material

Please note that in Table S1 data about the age of patient number 16 is missing.

Figures

Fig.4: it could be useful insert data also for healthy participants during NREM stage? Please better clarify what point C represents. In point B, it could be useful also present data for healthy participants?

Fig S2: point B, the figure seems to represent stimulation of the right parietal cortex rather than premotor cortex.

Fig S3: please better clarify the meaning of this figure with respect to the already presented correlation figures/analyses.

Reviewer #3 (Remarks to the Author):

Rosanova et al. probe changes in local and global brain dynamics evoked by non-invasive magnetic stimulation in patients with Unresponsive Wakefulness Syndrome. In brief, using TMS, the authors argue that the presence of local bistability in cortical activity - which is usually only present in NREM sleep – blocks a TMS-evoked burst of high frequency activity which in turns prevents global changes in cortical dynamics [1]. The resolution of this process, at least in one patient, is associated with clinical recovery.

The paper is interesting and the authors have exploited their access to a unique clinical population to probe general functional principles in the brain. The use of TMS as a probe is timely and the data analyses, as previously published [1], address the question. The comparison to NREM is also interesting. I think the paper is of sufficient potential interest and innovation for Nature Communications. However, I do have some reservations about the current ms.

1. I like the notion of bistability that the authors invoke but it does not seem that they have clearly demonstrated the same, here or in the previous paper [1] and as reviewed by them in a perspective [2]. In brief, a bistable system, as per is one in which an attractor landscape has two attractors, separated by a boundary (as per their [2]). The presence of a bistable system can be achieved by documenting bimodal and dwelling statistics, and formally showing that a dynamical model in a bistable regime is more likely, given the data, than one that has a single attractor – for an example, see [3,4]. I can't see how the demonstration of a non-responsive slow wave is clear evidence of bistability. I think the authors should endeavour to more formally test for bistability (which would be very interesting) or moderate their language (acceptable but less interesting).

2. I couldn't follow how the correlation analysis of Figure 3 supports the causality argument implicit in the phrase, "disrupt local causality". For a start, are the correlations within or across participants? (in [1] it seems they are within single subjects, here there is just one regression for each pair of variables). A within-subject analysis would be more convincing – i.e. in those single trials, when there is a slow wave, there is no burst, and conversely, there is a burst (even if it is diminished) if there is no slow wave. I am not convinced that a causal process is established if the regression is across participants.

3. The paper claims that the response is "regardless of stimulation site" although I only see BA6 and BA7 in the corresponding Figure 2A.

4. Discussion: Bistability, if present, has its neurobiological moderators/correlates but it is ultimately a system-wide phenomenon and I thought the Discussion might engage in such considerations.

Minor:

1. The text jumps from Figure 1A, to 2A and 2B, then back to the rest of Figure 1, then to Figure 3, without ever referring to 2C or 2D. It makes for a confusing read and I wonder if the authors could try for a more linear exposition of their panels in the main text.
2. Please provide a p-value for PCI (p7).

References:

1. A. Pigorini, S. Sarasso, P. Proserpio, C. Szymanski, G. Arnulfo, S. Casarotto, M. Fecchio, M. Rosanova, M. Mariotti, G. Lo Russo, J. M. Palva, L. Nobili, M. Massimini, Bistability breaks-off deterministic responses to intracortical stimulation during non-REM sleep, *Neuroimage* 112, 105–113 (2015).
2. M. V. Sanchez-Vives, M. Massimini, M. Mattia, Shaping the Default Activity Pattern of the

Cortical
Network, *Neuron* 94, 993–1001 (2017).

3. Freyer F, Roberts JA, Becker R, Robinson PA, Ritter P, Breakspear M (2012). A canonical model of multistability and scale-invariance in biological systems. *PLoS Computational Biology* 8: e1002634

4. Freyer F, Roberts JA, Becker R, Robinson PA, Ritter P, Breakspear M (2011). Dynamic mechanisms of multistability in the human alpha rhythm. *Journal of Neuroscience* 31: 6353-6361.

Response to the Reviewers' comments

Reviewer #1 (Remarks to the Author):

The current study focuses on unresponsive wakefulness syndrome (UWS); a condition where brain injured individuals do not show behavioral signs of consciousness, which the authors believe is due to an abnormal OFF-period response to transient cortical activation. Using transcranial magnetic stimulation in patients with UWS, the chief result is the cortical evoked response consists of a slow wave <4Hz and reduced power >20Hz, which resembles the EEG OFF-period response during NREM sleep, but not the response during wakefulness, in healthy individuals. In addition, EEG signal processing shows two measures, one of local causality and the other global complexity, decrease and plateau respectively during the evoked OFF period in patients with UWS. The study concludes sleep-like OFF periods could explain loss of consciousness in UWS.

This is an interesting study and presents new results of sleep-like EEG slow waves in patients with UWS. Strengths of the manuscript include good rationale, clever design using transcranial magnetic stimulation, clear presentation of results in text and figures, and interesting discussion that proposes two reasonable explanations for sleep-like OFF period in UWS. In spite of these strengths, it is curious why spontaneous slow wave were not included in the analysis, which could also indicate pathological bi-stability; the method for computing phase-locking, and how this is a measure of causal (neuronal) interactions, is unclear; and the neuronal correlates of EEG slow waves in UWS are yet to be identified, although the observations here present several hypotheses. These comments and other are discussed below.

Fig. 2 indicates spontaneous slow waves occur in patients UWS, similar to how slow waves occur during natural NREM sleep. Are the spontaneous slow waves associated with reduced high frequency power in UWS as in NREM sleep, and if it does, then could this correspond with inappropriate sleep-like OFF periods in UWS?

Following Reviewer's suggestions we performed analyses aimed at detecting individual slow waves in the spontaneous EEG of UWS patients. Specifically, we applied a well-established automatic slow waves detection algorithm (*Riedner et al., Sleep. 2007 Dec; 30(12):1643-57*) and applied time-frequency analysis to ascertain the presence of sleep-like OFF-periods associated with spontaneous slow waves.

Results are presented in the figure reported below (Supplementary Fig. 3) and confirm the Reviewer's intuition that even during behavioral wakefulness, i.e. eyes open, UWS patients may show spontaneous, "inappropriate" slow waves sharing the fundamental properties (i.e. high frequency suppression) of natural sleep slow waves (*Menicucci et al., Int J Psychophysiol. 2013 Aug;89(2):151-7; Piantoni et al., Int J Psychophysiol. 2013 Aug;89(2):252-8*).

Supplementary Figure 3. EEG recordings performed in awake UWS patients display a highly variable prevalence of spontaneous sleep-like slow waves and OFF-periods, which are invariably revealed by the TMS. In two representative UWS patients the prevalence of slow-wave activity was assessed in the background EEG according to a well-established automatic detection algorithm (6). (A) For both patients, on the left the topographical distribution of spontaneous slow wave density is shown. Grey traces in the cyan boxes represent individual spontaneous slow waves of the channel showing the maximum number of detections (waves/min), together with their average (black line), and the corresponding ERSP (bottom panel) (B) For both patients, single trials (grey traces) of the electrode showing the largest TMS-evoked potential (average responses are superimposed in black), together with the corresponding ERSP (bottom panel). Dashed vertical lines mark the occurrence of TMS. In the ERSP plots, the dashed horizontal lines indicate the 20 Hz frequency bin. This figure shows that i) slow waves (and the associated OFF-periods) can be immediately evident from the spontaneous EEG, and ii) TMS perturbations reveal the presence of OFF-periods even in those patients in whom slow waves are not prevalent.

However, the presence of spontaneous EEG slow waves is a highly variable finding both across and within UWS patients (see also Fig. 2B and C in the manuscript)(Casarotto et al., Ann Neurol. 2016 Nov;80(5):718-729; Kulkarni et al., J Clin Neurophysiol. 2007 Dec;24(6):433-7) further supporting the idea that the perturbational approach based on TMS/EEG recordings is suitable to best unveil the latent tendency of cortical circuits to fall into an OFF-period (due to adaptation mechanisms, see reply to Reviewer #3, comment 1) even when this tendency is not immediately evident from the spontaneous recordings.

We now comment upon this observation in the main text (Discussion - Page 12) as follows:

“In fact, TMS perturbations could reveal the presence of adaptation mechanisms and of the ensuing OFF-periods in all patients, regardless of their background EEG pattern (Supplementary Table 2), of the prevalence of spontaneously occurring slow waves (Supplementary Fig. 3), and of pre-stimulus ongoing activity (Fig. 2B, C and D).”

The methods for computing the phase-locking factor and how this measures local causal interactions is unclear. The text indicates an average Hilbert transform of the 8Hz highpass filtered evoked response was calculated, but no indication between which pairs of electrodes phase-locking was calculate or the spatial distribution of phase-locking values. Was it restricted only to the four adjacent recording sites with largest amplitude evoked response?

We thank the Reviewer for giving us the opportunity to clarify this point.

At odds with Phase locking Value (PLV), Phase Locking Factor (PLF) is calculated for single electrodes (rather than pairs) as an adimensional index (range 0–1) defined as the absolute value of the average of the Hilbert Transform across trials. To the extent that instantaneous PLF (i.e. the time-course of PLF) measures the coherence of the response to a perturbation across trials in a specific time-window, it can be used to quantify the duration of the deterministic effect of a given input. In fact, in a previous work from our group employing intracortical perturbations and

recordings (Pigorini et al., Neuroimage. 2015 May 15;112:105-113) we used PLF to demonstrate that the occurrence of cortical OFF-periods during NREM sleep can curtail the ability of a given cortical node to sustain long-lasting, deterministic responses to incoming inputs. In the present work, we use PLF to demonstrate that also pathological OFF-periods occurring in UWS patients are associated with a short lasting PLF. In order to clarify the meaning and use of PLF in the context of our manuscript we modified the Methods section as follows (Methods - Pages 20-21):

“Phase-locking factor. The impact of the OFF-periods on local causal interactions was assessed by means of broadband (> 8 Hz) phase-locking factor (PLF; 85). PLF can be calculated for every single electrode as an adimensional index (range 0–1) defined as the absolute value of the average of the Hilbert Transform across trials. To the extent that instantaneous PLF (i.e. the time-course of PLF) measures the coherence of the response to a perturbation across trials in a specific time-window, it can be used to quantify the duration of the deterministic effect of a given input 10,85. Here, for each EEG channel i (1 to 60), single trials were high-pass filtered above 8 Hz (third order Butterworth filter) and PLF was computed as the absolute value of the average of the Hilbert Transform of all single trials. Assuming a Rayleigh distribution of the baseline values from -500 ms to -100 ms, PLF time points that were not significantly different from baseline ($\alpha < 0.01$) were set to zero. For each channel i , the latest significant PLF time point was identified and labelled as $\max \text{PLF}_t(i)$. Finally, $\max \text{PLF}_t$ (timing of the last significant time point of phase-locking) was calculated as the average of $\max \text{PLF}_t(i)$ over the four channels closer to the stimulation site (Supplementary Fig. 1D).”

At odds with our previous intracranial observations, in scalp EEG data the evoked OFF-periods were clearly detectable and significant only close to the stimulation site. For this reason, in order to highlight the temporal relationship between OFF-periods and PLF duration, we limited our analysis and computed PLF on the four electrodes under the TMS coil.

However, in order to fully address the Reviewer’s comment, we averaged the $\max \text{PLF}_t$ (timing of the last significant timepoint of the time-resolved PLF) across all electrodes and compared the results across conditions (UWS, HW, HS). Results are presented below for the Reviewer’s benefit.

Parietal Stimulation			Frontal Stimulation
Wilcoxon ranksum test (P)		Wilcoxon signrank test (P)	Wilcoxon ranksum test (P)
HW(20) vs UWS(14)	UWS(14) vs HS(8)	HW(8) vs HS(8)	HW(20) vs UWS(10)
max PLFt	2.227*10 ⁻⁴	0.322	0.048

Finally, we replaced a reference regarding PLF with a more relevant one. Specifically the paper by Palva and colleagues titled *Phase synchrony among neuronal oscillations in the human cortex* (Palva et al., J Neurosci. 2005 Apr 13;25(15):3962-72) has been replaced with the paper authored by Palva and

colleagues titled *Early neural correlates of conscious somatosensory perception* (Palva et al., J Neurosci. 2005 May 25;25(21):5248-58). We apologize for the confusion.

In regard to the perturbational complexity index (PCI), have there been any studies on the neuronal basis of PCI? Other than the behavioral correlates of PCI it would be helpful to understand the neurophysiology that correlate with PCI values.

A recent study by D'Andola and colleagues (*D'Andola et al., Cereb Cortex. 2018 Jul 1;28(7):2233-2242*), quoted in the present manuscript, represents a first step towards the understanding of the neuronal underpinnings of PCI. Specifically, these authors calculated perturbational complexity in an *in-vitro* model (electrical perturbations of ferret cortical slices), during pharmacological manipulations aimed at modifying spontaneous activity in active cortical networks. Results showed that OFF-periods in cortical slices are associated with an early break-off of phase-locking of the response and with low PCI values. A reduction of bistability in pharmacologically-induced awake-like states resulted in sustained phase-locking and higher PCI values.

The present study parallels this microscale results *in-vitro* with observations at the bedside of patients. Clearly, a solid mechanistic link between these two levels would require further exploration at the mesoscale level by means of intracranial human and animal recordings. Although this explicit link is still missing, it is worth noting that *in-vitro* cortical slices reproduce the fundamental behavior of *in-vivo* cortical circuits after deafferentation (*Timofeev et al., Cereb Cortex. 2000 Dec;10(12):1185-99*), a common consequence of brain injury. This point is discussed in the revised version of the Discussion section (Discussion - Page 14) that now reads:

“As a proof of principle, a recent microscale study employing electrical stimulation and recordings in isolated cortical slices showed that phase-locking and complex causal interactions, as assessed by an adapted version of PCI, could be effectively restored by pharmacological interventions that reduce bistability and increase cortico-cortical excitability 69. This microscale finding further suggests a causal link between cortical bistability and complexity and may have translational implications since brain slices can be considered a simplified model of the electrophysiological state of the cerebral cortex under conditions of severe deafferentation.”

Changes in PLF values during the evoke response illustrated in the figure appear more variable than what is described in the text. While it is clear the latest significant PLF value occur later during the healthy awake condition than UWS or NREM sleep, what about the first and other non-significant PLF time point before the latest time point?

In healthy wakefulness PLF is indeed variable in the sense that it is fluctuating above and below the threshold for statistical significance after the TMS pulse (as shown in Fig. 1 for a representative subject). These fluctuations around the statistical significance threshold have been already observed in *in-vitro* recordings (*D'Andola et al., Cereb Cortex. 2018 Jul 1;28(7):2233-2242*) as well as in intracerebral recordings in humans (*Pigorini et al., Neuroimage. 2015 May 15;112:105-113*) and interpreted as reflecting both intrinsic neuronal properties as well as feedback from neighboring elements. Since PCI necessarily builds upon local and global deterministic effects, we found appropriate to correlate its time-course with the end of the overall deterministic interactions rather than with the timing of their first transient dip below significance level (as shown in Fig. 1A) in order to avoid a coarse underestimation of the duration of actual causal effects.

In Fig. 4B, why is the ERSP >20Hz increasing before, and maximal at the onset of TMS?

The standard method to assess the modulation of power across different EEG frequency bands with a good time resolution is the Event Related Spectral Perturbation (ERSP) algorithm as

implemented in the EEGLAB toolbox. Calculation of ERSP involves time-frequency decomposition based on Morlet wavelets (*Delorme and Makeig*, J Neurosci Methods. 2004 Mar 15;134(1):9-21) sliding over time windows. The duration of these time windows varies across frequency bins as their duration is expressed in oscillation cycles. Here, as in previous works (*Pigorini et al.*, Neuroimage. 2015 May 15;112:105-113; *Rosanova et al.*, J Neurosci. 2009 Jun 17;29(24):7679-85; *Ferrarelli et al.*, Arch Gen Psychiatry. 2012 Aug;69(8):766-74; *Fecchio et al.*, PLoS One. 2017 Sep 14;12(9):e0184910) we used a fixed number of wavelet cycles (3.5) in order to achieve an optimal trade-off between frequency and time resolution. This results in an artificial backward spread of power, maximal in the low frequency range, with an onset at \sim -50 ms and explains the profile illustrated in Fig. 4B and in other time-frequency illustrations.

Also, why do the PLF values fall to zero at max PLFt, and PCI value plateau at 0.12 at max PCIIt? Is due to the methodology or reflect physiological brain activity? This should be explained or at least noted in the legend.

We defined max PLFt as the last significant point of the PLF (assuming a Rayleigh distribution for the baseline values). In the previous version of the manuscript, we specified having set to zero all the non-significant PLF time points. For the sake of clarity, we now modified Fig. 4 and Supplementary Fig. 1 and present the actual PLF values with the statistical threshold shaded in grey. We apologize for the confusion.

As for the time-course of PCI, its values are computed based on the study of Casali and colleagues (*Casali et al.*, Sci Transl Med. 2013 Aug 14;5(198):198ra105). As such, max PCIIt merely reflects the last time point at which PCI (calculated over a time window of 8-300 ms) shows increments.

The manuscript includes discussion of “...neuronal OFF-periods...” and “...neuronal events...” in explaining the some of the current findings, but it is not known what is happening at the level of neurons, unlike some unit studies of NREM slow waves in presurgical patients (e.g. Nir et al. 2011).

We fully agree with the Reviewer. What we can actually observe through EEG scalp recordings is just a proxy of neuronal activity. Ideally, one should record, as in the study by Nir and colleagues quoted by the Reviewer, spiking activity from single cortical units during spontaneous and evoked slow wave activity in UWS patients. Since this is not feasible in UWS patients, we adopted an accepted procedure to detect putative OFF-periods based on non-invasive EEG recordings. Previous animal studies combining intracellular, multiunit and local field potential (LFP) (*Mukowski et al.*, Cereb Cortex. 2007 Feb;17(2):400-14) demonstrated a reliable relationship between neuronal down states (as recorded intracellularly) and the suppression of high-frequency oscillations (>20 Hz) in the extracellular LFP, which have been called cortical OFF-periods (see figure below).

Based on these animal studies and on subsequent intracranial recordings during sleep in humans (Cash et al., *Science*. 2009 May 22;324(5930):1084-7; Valderrama et al., *PLoS One*. 2012;7(4):e33477), a similar procedure has been implemented and validated that detects the OFF-periods associated with slow waves at the scalp EEG level (Piantoni et al., *Int J Psychophysiol*. 2013 Aug;89(2):252-8; Menicucci et al., *Int J Psychophysiol*. 2013 Aug;89(2):151-7). These works demonstrated a tight relationship between the phase of the slow oscillations (and K-complexes) and the alternation between up and down states as reflected by high frequency modulations. Crucially, here, by applying the same methods to TMS-evoked scalp EEG responses, we replicated the same findings obtained with intracranial stimulations and recordings during NREM sleep in humans (Pigorini et al., *Neuroimage*. 2015 May 15;112:105-113).

We now clarified the relationship between neuronal and extracellular detected events by adding all the relevant references at the beginning of the Results section (Results - Page 5).

Moreover, we replaced in the text all the instances referring to “down states” (which refers to a specific intracellular event) with OFF-periods in order to stress the indirect, extracellular nature (high-frequency suppression) of our detections.

Other comments:

Unlike EEG, MRI, NREM sleep, and maybe TMS, extensive use of uncommon abbreviations interrupts the flow of sentences and hinders readability, e.g. TMP, TEP, PCI, SPES, PLF, HFp, max SWa, max PLFt, max SHFt, max SHFp, PCIt, and CRS-R.

As suggested by the Reviewer, in the revised version of the manuscript we have reduced the number of abbreviations (e.g. TEP, SPES, LFP, DAI and NBS have been removed). Moreover, where needed, we have explicitly detailed the meaning of acronyms.

In Table S1, what does severe, moderate, and mild mean in EEG category?

These categories refer to a classification of the level of abnormality of the spontaneous EEG recorded in patients affected by disorders of consciousness after anoxic or traumatic injury based on visual criteria (Forgacs et al., *Ann Neurol*. 2014 Dec;76(6):869-79) now also quoted in the revised caption of Supplementary Table 2 (formerly Table S1).

Should spell out ERSP in first appearance in the manuscript.

ERPS is spelled out in the first appearance in the manuscript (Legend of Fig. 1 and Methods section)

Reviewer #2 (Remarks to the Author):

Authors here present an interesting study about the effect of perturbation, performed by TMS, on the brain responses of patients with unresponsive wakefulness syndrome (UWS). They demonstrate that these patients, when awake, are characterized by states that are very close to those physiologically observed in healthy participants during non-REM sleep. Thus, they suggest that this pathological sleep-like bistability with loss of causality and complexity, in the brain of UWS patients (also in cortical regions apparently spared from injuries), could be related with their difficult in recovering full consciousness. Turning this pathological condition toward higher levels of cortical excitability could be useful for recovering of these patients.

The manuscript is interesting, and could give some new and useful information in the field. However, in my opinion, it could be improved in some sections. Thus, I have some major and minor concerns that should be managed by Authors.

Major concerns

Abstract: The present version of the abstract could be improved by using a more “structured” model. Now, it seems more like a piece of an “introduction” section of a manuscript. Please give some adjunctive information (especially about groups, methods and results), obviously compatibly with editorial guidelines of the Journal (e.g. word limit, “style” of abstracts, and so on...).

The Reviewer can find below a new version of the abstract, which we have included in the revised manuscript and which contains some additional information. However, it seems that this new abstract largely exceeds (by c.ca 50 words) the word count limit explicitly stated in the Nature Communications Guide to Authors. We leave to the Editor the decision of whether to include or not this new version of the abstract in the revised version of the manuscript.

“Unresponsiveness Wakefulness Syndrome (UWS) patients may retain intact portions of the thalamocortical system that are spontaneously active and responsive to sensory stimuli but fail to engage in complex causal interactions, resulting in loss of consciousness. We show that loss of brain complexity after severe injuries is due to a pathological tendency of cortical circuits to fall into a period of silence (OFF-period) upon receiving an input. This basic mechanism is known as cortical bistability and has been thoroughly investigated during sleep. Spectral and phase domain analysis of EEG responses to transcranial magnetic stimulation in UWS patients (n=16) revealed the occurrence of OFF-periods that were never present in healthy awake individuals (n=20). These pathological OFF-periods were similar to the ones evoked in the cortex of healthy sleeping subjects (n=8) and were detected in all targeted cortical areas (both frontal and parietal bilaterally). Most important, these OFF-periods significantly impaired local causal interactions and prevented the build-up of global complex interactions in the brain of UWS patients. The present findings draw a direct link between well-studied and potentially reversible local events (OFF-periods) and global brain dynamics that are relevant for pathological loss and recovery of consciousness.

Results/Methods

It is clear how you decided to stimulate premotor or parietal cortex but it is not clear how and when you decided to stimulate left and/or right hemisphere; moreover, it is not clear in which patients you stimulated left and/or right; moreover, when considering also healthy participants this could result in additional TMS/EEG measurements (more than 72?), but, again, it is not clear when you stimulated left/right premotor/parietal cortex, in healthy subjects, during awake and/or during sleep (in the same day? Really?). Please

clarify (further tables in the supplementary material could help?). Furthermore, please give further detail about how you decided to manage data: i.e., in healthy participants, data obtained from left/right premotor/parietal cortex (in the different conditions) were always fully comparable with respect to the same measurements (perhaps partial) obtained from patients? Moreover, it is not always clear if data obtained from the different cortices were, at the end, merged for analyses or not. Please clarify.

In both healthy subjects and UWS patients we targeted TMS over frontal cortex (Brodmann Area 6) and parietal cortex (Brodmann Area 7). Specifically, in each healthy subject we have stimulated two cortical areas either on the left or on the right side, counterbalanced across individuals. This information is now reported in a new table (Supplementary Table 3) and better specified in the Methods section. In line with previous works (*Harquel et al., Neuroimage. 2016 Jul 15;135:115-24; Casali et al., Sci Transl Med. 2013 Aug 14;5(198):198ra105*), we found no significant difference between left and right stimulations.

TMS/EEG recordings during sleep in healthy subjects were performed during nighttime on the same day of the parietal stimulation during wakefulness. Specifically, within the same recording session we acquired the wakefulness measurements (Brodmann Area 6 and Brodmann Area 7) before lights off and the sleep measurement (Brodmann Area 7) as soon as the participant reached a stable N3 sleep stage.

In patients, we aimed at the same stimulation targets (namely, Brodmann Area 6 and Brodmann Area 7) as in healthy participants, this time bilaterally. This was required since in brain-injured patients not all sites can be accessed due to lesions, whose stimulation fails to elicit any significant cortical response (see *Gosseries et al., Brain Stimul. 2015 Jan-Feb;8(1):142-9*), and extracranial derivation for liquor drainage.

We agree with the Reviewer on the need to better specify stimulation targets in the revised version of the manuscript. For this reason, we now report L/R in Supplementary Table 2 (formerly Table S1) and we add a new table reporting stimulation targets (plus demographics) also for healthy subject (Supplementary Table 3).

Finally, in order to clarify which analysis included data merged across stimulation sites, we modified Fig. 3 and Fig. 4 and indicated with different markers the data pertaining to stimulation performed over Brodmann Area 6 and Brodmann Area 7. For all the other figures we now clearly indicate in the figure caption the relative cortical targets.

I am not always convinced that using Bonferroni correction is really informative, it depends from the context. In this case, I think that it could be not used because this study is more close to a “proof of evidence”, that will deserve further confirmation. Moreover, you give not information about the “corrected” statistical thresholds. Thus, I prefer that you indicate the “real” p-values (and not simply $p < 0.05$ or $p < 0.01$): this could be more informative about the real (and evident) effects. Similarly, information about tests used (i.e. when non-parametric ANOVA was used or when Wilcoxon/Mann-Whitney was used) should be given in a more systematic way when presenting results (or in the methods section).

We fully agree with the Reviewer. Accordingly, in the revised version of the Results section we reported all the real P values in both the new tables (Table 1 and Supplementary Table 1) indicating the statistical tests used for each comparison. In addition, following the Reviewer’s suggestion, we indicated the different statistical tests used in the paragraph “Statistical Analysis” of the Methods section that now reads as follows:

“Group analyses were performed in MATLAB by using Wilcoxon ranksum test and Wilcoxon signrank test where appropriate (see Table 1 and Supplementary Table 1 for details).”

Methods

Please insert a reference when you speak about the hallmarks of cortical OFF-periods, and when you present the calculation of the duration of the broadband PLF.

We thank the Reviewer for pointing out these issues. Concerning the hallmarks of cortical OFF-periods in the new version of the manuscript we have better defined the concept of OFF-period based on previous literature (see also response to Reviewer #1). Specifically, we included all the relevant references in the revised version of the Results section.

On the other hand, the procedure to calculate the duration of the broadband PLF (max PLFt) is the same used in a recent paper from our lab (Pigorini et al., Neuroimage. 2015 May 15;112:105-113) and is derived from previous literature (Palva et al., J Neurosci. 2005 May 25;25(21):5248-58). In the revised version of the Methods section, this reference is now quoted (see also response to Reviewer #1).

Here, you say that only eight healthy participants were measured during NREM sleep: please give further information (age, sex, etc.).

Following the Reviewer’s suggestion, in the revised version of the manuscript we report this information in Supplementary Table 3.

Which could be the statistical implications of using a “sub-group” of healthy participants when comparing data with UWS and awake healthy subjects?

Given this and a similar comment from the same Reviewer (see above), we realize that performing a non-parametric ANOVA (as in the original version of the manuscript) including a within-subject comparison (HW vs HS) with unbalanced numerosity between samples (20 vs 8) may not represent the correct way to statistically handle our data.

Therefore, we modified our statistical approach and tested separately the different effects using the appropriate statistical tests and number of cases. Specifically, we applied Wilcoxon ranksum test when comparing HW (n=20) to UWS (n=14) as well as when comparing HS (n=8) to UWS (n=14). On the other hand, we applied Wilcoxon signrank test when comparing HW (n=8) to HS (n=8).

As such, we modified the paragraph “Statistical Analysis” of the Methods section that now reads as follows:

“Group analyses were performed in MATLAB by using Wilcoxon ranksum test and Wilcoxon signrank test where appropriate (see Table 1 and Supplementary Table 1 for details).”

Also, we included two new tables (Table 1 and Supplementary Table 1) reporting details regarding the applied tests, the sample size and the significance values for each comparison between conditions (HW, HS, UWS).

Also, we modified Fig. 1 and Supplementary Fig. 2 accordingly. Specifically, in the new version of Fig. 1D and Supplementary Fig. 2C, bar graphs have been replaced by boxplots (individual values, median and interquartile range) for max SWA, HFp and max PLFt calculated in all groups of study participants including the subgroups of HW.

Finally, we have revised the Results section accordingly.

From the last paragraph at p. 16, it seems that, even if the same lobe was stimulated, the cortical targets may be, at the end, slightly different in UWS and healthy participants. Please clarify implications of this.

We thank the Reviewer for allowing us to clarify this potentially relevant issue. First, TMS targets in UWS patients were constrained by the presence of cortical lesions therefore not allowing for a specific, consistent selection of TMS targets across patients. Nonetheless, TMS, as revealed by modeling studies (*Opitz et al., Neuroimage. 2011 Oct 1;58(3):849-59; Thielscher et al., Neuroimage. 2011 Jan 1;54(1):234-43*), is not pinpointing specific sets of cortical neurons, but rather it coarsely perturb large patches of the cerebral cortex.

In this and previous works we exploit this property and we aimed at strongly activating macroscopic cytoarchitecturally defined cortical patches, i.e. Brodmann areas, each including multiple gyri with the goal of assessing general impulse-response properties of the brain.

In this perspective, it is conceivable to assume that the impact of slightly different TMS targeting across UWS patients would be negligible also given the fact that EEG response to TMS (slow waves associated with OFF-periods, the main aim of this work) is highly reproducible independently of the stimulated Brodmann area.

Please give further information about how you decided to set intensity of stimulation.

As in previous works (*Rosanova et al., J Neurosci. 2009 Jun 17;29(24):7679-85; Casarotto et al., Ann Neurol. 2016 Nov;80(5):718-729*), based on the neuronavigation system the intensity of the TMS-induced electric field was set always above 120 V/m both in UWS patients and healthy controls. The cutoff of 120 V/m has been chosen stemming from results of previous studies (*Casali et al., Neuroimage. 2010 Jan 15;49(2):1459-68; Casarotto et al., PLoS One. 2010 Apr 22;5(4):e10281*) in which we show that this intensity of the induced electric field allow for a significant and reproducible EEG response to TMS. In order to detail this methodological aspect, the relevant paragraph of the Methods section now reads (Methods - Page 17):

“The intensity of the TMS-induced electric field was always set above 120 V/m based on the neuronavigation system. The intensity of 120 V/m has been shown to generate significant and reproducible TMS-evoked EEG potentials 79,80. Overall, the TMS-induced electric field was comparable between UWS patients (124.8 ± 9.8 V/m and 138.3 ± 12.9 V/m, $\text{mean} \pm \text{SEM}$, for parietal and frontal stimulation, respectively) and awake healthy subjects (132.9 ± 4.8 V/m and 133.9 ± 5.5 V/m, Wilcoxon ranksum test, $P = 0.452$ and $P = 0.813$, for parietal and frontal stimulation, respectively). In healthy subjects who underwent TMS/EEG measurements both during wakefulness and sleep, the same stimulation parameters were applied by means of the Navigated Brain Stimulation system.”

Why did you use a disproportionate number of males/females in the two groups? Could this have an effect in terms of data “noise”? Similarly, it could be useful to have some additional data about the age of every healthy participant.

Thanks to the Reviewer’s question we realized that the number of female (n=14) healthy volunteers reported in the old version of the manuscript was not the correct one. In fact, 14 is the total number of females considering the two groups of study participants, i.e. healthy volunteers and UWS patients together. Indeed, the number of females in the group of UWS patients and healthy volunteers is 6 and 8, respectively. In the revised version of the manuscript, we reported the number of female separately for the two groups in Supplementary Table 2 and Supplementary Table 3.

Correlations were performed (separately) in every group or by “merging” data of every group?

Correlations were performed for the UWS patients group only. With respect to sleep correlations, these were thoroughly described in previous works (see also our response to the Reviewer’s comment about Fig. 4) and, as such, were out of the main aim of this work.

As correlations included merged data from both Brodmann Area 6 and Brodmann Area 7, we modified Fig. 3 and Fig. 4 and indicated with different markers the data pertaining to each stimulation site.

Discussion

Discussion is well written and structured, but it seems that (sometimes) speculations are very “invasive” (e.g. toward the end of p. 11 or in the first four/five rows of p. 12) Please try to avoid this.

We understand the Reviewer’s concern and modified the revised version of the Discussion section accordingly. As an example, we removed the explicit link to theories on consciousness previously included at the beginning of Page 12. Also, we shortened the long list of questions previously included in the second paragraph of Page 14. Finally, we were not able to shorten the discussion at Page 11 as it seems instrumental for addressing concern 1 of Reviewer 3.

In all cases, we hope that readability has now improved and we thank the Reviewer for his/her suggestion.

Minor concerns:

Introduction section

The introduction is informative and concise. However, please better define in this section the concept of low-complexity UWS patients (p. 4), also for less expert readers.

The introduction section of the revised manuscript reports a more exhaustive description of low-complexity UWS patients and now reads:

“However, in most cases, the EEG response to TMS is simple and stereotypical, as assessed by the perturbational complexity index (PCI): in these patients, identified as “low-complexity” UWS, TMS elicits a strong initial activation, which fails to evolve into complex patterns of interactions 7.”

Results

Just a curiosity: it could be useful/possible to correlate also SWa/maxSHFp with maxSHFt/PLFt?

Following the Reviewer’s suggestion, we computed the correlation between the two proportions (max SWa/maxSHFp vs maxSHFt/PLFt). Only for the Reviewer’s benefit, we attach the figure displaying the correlation from which is evident that the two proportions do not correlate significantly.

Besides this negative finding, we were not able to provide any compelling neurobiological explanation for this correlation. However, in case the Reviewer would like to suggest any

interpretation we are open to consider it.

Methods

Here you write that interstimulus interval was 2000-2300 ms. In figures you indicated 5000-5300 ms. Please clarify.

In this work, the use of an interstimulus interval of 5000-5300 represents a particular case limited to results presented in Fig. 2 for a representative patient in which an additional stimulation session with this specific interstimulus interval was performed. Specifically, in this case, the longer interstimulus interval was chosen to maximize the time window of the baseline EEG before the TMS stimuli thus showing more clearly the presence/absence of spontaneous slow waves in the ongoing activity during the stimulation sessions (the purpose of Fig. 2). For all the other results presented in the manuscript the applied interstimulus interval is 2000-2300.

Supplementary material

Please note that in Table S1 data about the age of patient number 16 is missing.

We thank the Reviewer for pointing out this missing information. Gender and age for Patient 16 have now been added in the Supplementary Table 2 (formerly Table S1).

Figures

Fig.4: it could be useful insert data also for healthy participants during NREM stage?

The main point of Fig. 4 is to focus on a condition (UWS) in which subjects are awake with eyes open but show the occurrence of OFF-periods and low complexity in response to TMS as compared to healthy wakefulness. Time courses of complexity during healthy NREM sleep were reported in the study by Casali and colleagues (*Casali et al., Sci Transl Med. 2013 Aug 14;5(198):198ra105*) and correlations between OFF periods and PLF were thoroughly described by a previous intracranial study (*Pigorini et al., Neuroimage. 2015 May 15;112:105-113*). Thus, in Fig. 4 we focused our exploration on the relationships between OFF-periods, short PLF, and low complexity in UWS and limited our exploration to the confirmation their occurrence during NREM sleep only in a subset of healthy subjects (n=8) as reported in Fig. 1.

Please better clarify what point C represents.

The correlations reported in Panel C of Fig. 4 shows that the occurrence of an OFF-period and the drop of the PLF prevents the build-up of PCI in UWS patients.

In point B, it could be useful also present data for healthy participants?

Fig. 4B showcases the results for UWS patients and illustrates, for a representative case, that the OFF-period occurs at the same time with the drop of PLF and the plateau of PCI growth over time. As the OFF-periods are never present in healthy awake subjects (see Fig. 1 and Supplementary Fig. 2) these results are not part of Fig. 4.

Fig S2: point B, the figure seems to represent stimulation of the right parietal cortex rather than premotor cortex.

We thank the Reviewer for pointing out this mistake. Supplementary Fig. 2 has been now corrected in the revised version of the manuscript.

Fig S3: please better clarify the meaning of this figure with respect to the already presented correlation figures/analyses.

Supplementary Fig. 3 refers to all TMS/EEG measurements that we performed in patients including additional targets and stimulation intensities other than those systematically explored (other than the ones reported in the Methods section and in the main results). The point of this figure was to show that correlations are robust and that can be generalized across different targets and stimulation intensities. However, since we realized that this figure is not essential for the flow of the paper, we have now replaced it with a new figure (Supplementary Fig. 3), which addresses specific comments to the concerns of Reviewer #1 and Reviewer #3.

§ § § § § § § § § § §

Reviewer #3 (Remarks to the Author):

Rosanova et al. probe changes in local and global brain dynamics evoked by non-invasive magnetic stimulation in patients with Unresponsive Wakefulness Syndrome. In brief, using TMS, the authors argue that the presence of local bistability in cortical activity - which is usually only present in NREM sleep – blocks a TMS-evoked burst of high frequency activity which in turns prevents global changes in cortical dynamics [1]. The resolution of this process, at least in one patient, is associated with clinical recovery.

The paper is interesting and the authors have exploited their access to a unique clinical population to probe general functional principles in the brain. The use of TMS as a probe is timely and the data analyses, as previously published [1], address the question. The comparison to NREM is also interesting. I think the paper is of sufficient potential interest and innovation for Nature Communications. However, I do have some reservations about the current ms.

1. I like the notion of bistability that the authors invoke but it does not seem that they have clearly demonstrated the same, here or in the previous paper [1] and as reviewed by them in a perspective [2]. In brief, a bistable system, as per is one in which an attractor landscape has two attractors, separated by a boundary (as per their [2]). The presence of a bistable system can be achieved by documenting bimodal and dwelling statistics, and formally showing that a dynamical model in a bistable regime is more likely, given the data, than one that has a single attractor – for an example, see [3,4]. I can't see how the demonstration of a non-responsive slow wave is clear evidence of bistability. I think the authors should endeavour to more formally test for bistability (which would be very interesting) or moderate their language (acceptable but less interesting).

In the sleep field, cortical bistability indicates the latent tendency of cortical neurons to fall into an OFF-period due to adaptation currents in response to a transient increase of activity. Classically, this phenomenon has been studied under the extreme condition of deep anesthesia, during which cortical circuits enter a regime of regular alternation between active (up) and silent (down) states at a rate of about 1 Hz (the slow oscillation)(*Steriade et al., J Neurosci. 1993 Aug;13(8):3252-65; Steriade et al., J Neurosci. 1993 Aug;13(8):3266-83*). We understand that this rhythmic alternation, when analyzed in the framework of dynamical systems, is underlined by a single attractor (limit-cycle).

However, in the present work, we are analyzing a different, less stereotypical pattern, more similar to natural sleep (*Steriade et al., J Neurophysiol. 2001 May;85(5):1969-85; Timofeev et al., Proc Natl Acad Sci U S A. 2001 Feb 13;98(4):1924-9*), in which cortical circuits may linger either

in a state of tonic firing or, at times, enter in a slow oscillatory state whereby small transient increases in firing rate are inescapably followed by a silent period.

This situation is exemplified in a new figure (Supplementary Fig. 3) that has been included in the revised version of the manuscript (also see the response to Reviewer#1) as well as in Fig. 2 of the original manuscript. In more details, the analysis of the spontaneous activity in Supplementary Fig. 3 shows two patients in which cortical activity (as reflected in the EEG), displays periods of slow waves (associated with the alternation between strong increases of activity and suppressions, which were not computable in Patient 3 due to the low number of events). The occurrence of these events is covering variable stretches of the recording (up to 25 slow waves/min) intermingled to periods in which the EEG is stable and levels of activity are intermediate.

Supplementary Figure 3. EEG recordings performed in awake UWS patients display an highly variable prevalence of spontaneous sleep-like slow waves and OFF-periods, which are invariably revealed by the TMS. In two representative UWS patients the prevalence of slow-wave activity was assessed in the background EEG according to a well-established automatic detection algorithm (6). (A) For both patients, on the left the topographical distribution of spontaneous slow wave density is shown. Grey traces in the cyan boxes represent individual spontaneous slow waves in the channel showing the maximum number of detections (waves/min), together with their average (black line), and the corresponding ERSP (bottom panel) (B) For both patients, single trials (grey traces) of the electrode showing the largest TMS-evoked potential (average responses are superimposed in black), together with the corresponding ERSP (bottom panel). Dashed vertical lines mark the occurrence of TMS. In the ERSP plots, the dashed horizontal lines indicate the 20 Hz frequency bin. This figure shows that i) slow waves (and the associated OFF-periods) can be immediately evident from the spontaneous EEG, and ii) TMS perturbations reveal the presence of OFF-periods even in those patients in whom slow waves are not prevalent.

This bimodality is clear in the figure below (Fig. 2B and C in the manuscript) where the two patterns are shown within the same recording.

We think that this condition, in which these two states (tonic firing and slow oscillatory states) coexist, may indeed be referred to as bistable in the framework of dynamical systems.

Specifically, we argue that this regime is described by the blue area of the codimension-two bifurcation diagram (see arrow below) where low firing and slow oscillatory states are both stable attractors (figure modified from *Mattia and Sanchez-Vives*, Cogn Neurodyn. 2012 Jun;6(3):239-50).

This is a condition in which adaptation levels are such that, although a given level of tonic firing can be sustained, even small, transient activations are enough to precipitate the system in a stereotypical cycle encompassing an OFF-period, thus strongly limiting the system's capability for complex patterns of activity.

In this scenario, depending on the frequency of excitatory inputs, periods of spontaneous slow waves can be frequent (see above Supplementary Fig. 3 upper left corner) or more sporadic (see above Supplementary Fig. 3 bottom left corner) and thus the presence of adaptation mechanisms may not be promptly detected by purely observational perspective unless the system dynamics is recorded for long time.

Experimentally TMS, by transiently and artificially increasing activity levels, promptly reveals the presence of OFF-periods and of underlying adaptation currents even when those are not prominent in the ongoing spontaneous activity.

In view of the above, we think that it might be possible to draw an explicit link between the neurophysiological definition of cortical bistability (as revealed by TMS) and the classical notion of bistability in dynamical systems.

Formally testing this relationship by means of simulations is indeed extremely interesting and represents the topic under scrutiny in a future manuscript in preparation. Below, for the Reviewer's benefit, we show a neural system with high adaptation (left) which, during the period of observation, rests in a tonic firing state intermingled by spontaneous sequences of OFF-periods, but that promptly enters a slow oscillation associated with a down state each time it is perturbed, thus loosely representing the case of the patients in Supplementary Fig. 3. On the contrary, the figure on the right shows a situation of low adaptation in which the system rests in a

tonic firing state and responds to perturbations with variable increases in firing but never plunging into an OFF-period, possibly reflecting the condition of an awake healthy subject.

Trajectories describing the firing rate of the excitatory population as a function of the spike-frequency adaptation (SFA), and the corresponding time series, for high (left) and low (right) values of g , which accounts for the intensity of SFA. Simulated networks are composed of 5000 leaky integrate-and-fire neurons (80% excitatory and 20% inhibitory neurons as in Mattia, Sanchez-Vives, Cogn Neurodyn. 2012 Jun;6(3):239-50). SFA applies only to excitatory neurons, and it is such that at each emitted spike the adaptation variable is increased by a multiplying parameter g . SFA decays towards its resting value (0) with a time constant of 500 ms. External inputs (see yellow arrows), with fixed intensity and duration, is applied only to excitatory neurons.

Left, the system displays a stable tonic firing attractor as well as an external orbit characterized by an initial transient increase in firing rate followed by a proportional increase in SFA. This, in turn leads to a massive drop (up to a complete silencing) in firing rate, typical of a slow oscillatory state. The alternation between stable and limit-cycle attractors occurs spontaneously in the network due to finite-size effects, as illustrated in the first half of the time series. Crucially, perturbations (yellow arrows) always bring the network in limit-cycle attractor for at least one cycle, thus systematically revealing underlying adaptation. This dynamics qualitatively resembles recordings in UWS patients.

Right, the system displays a fully stable tonic firing attractor. In this case, due to a lower g , the same perturbations do not lead the system into an OFF-period. This dynamics qualitatively resembles recordings in healthy awake subjects.

To summarize the above, we believe that our experimental observations are compatible with the notion of bistability as the bimodal alternation between coexisting, stable states (tonic firing and slow waves) identified by two attractors. In this context, the transition between the two attractors is enabled by the strong influence of adaptation mechanisms that are promptly revealed by an external perturbation (TMS). This is crucial because adaptation mechanisms and the ensuing OFF-periods seem to prevent the build-up of complex interactions.

Although we do not include the results of the simulations illustrated above, we capitalize on the Reviewer's comment and the ensuing analysis to disambiguate and better qualify the notion of bistability throughout the text. In the revised version of the manuscript, we now explicitly define the neurophysiological notion of cortical bistability from the outset as the latent tendency of cortical neurons to fall into an OFF-period due to adaptation currents in response to a transient increase of activity. Accordingly, the revised version of the Introduction section (Introduction - Page 4) now reads as follows:

“This intrinsic propensity of cortical neurons to fall into OFF-periods has been thoroughly studied in the realm of sleep physiology across species and models and is referred to as cortical bistability 12,13. In silico, in vitro as well as in vivo animal models suggests that cortical bistability is due to adaptation mechanisms, such as activity-dependent K⁺ currents 14,15 as well as active inhibition 16,17.”

In addition, in order to improve readability and avoid confusion between the neurophysiological and the system-dynamic definition of bistability, we removed the term “bistable dynamics” and similar wordings when referring to our experimental observations from the main text.

At the same time, we hint in the revised Discussion section to the interesting link between the neurophysiological definition and the formal definition of bistability in the dynamical system framework, which will be explored in an upcoming work. Accordingly, we added a paragraph to the Discussion (see also response 4 to the same Reviewer), which now reads (Discussion - Pages 11-12):

“Overall, different mechanisms, alone or in combination, may engender a tendency towards cortical bistability in brain-injured patients, as also reflected by the presence of slow waves in their spontaneous waking EEG (Supplementary Fig. 3) 50–54. While the relative contribution of these mechanisms is difficult to disentangle, it is worth noting that all the above mechanisms can be effectively engaged by a cortical perturbation. For example, a direct cortical hit with TMS may (i) trigger activity-dependent K⁺ currents and an OFF-period, if K⁺ channels are de-inactivated; (ii) massively recruit local inhibitory circuits leading to an OFF-period, if the excitation-inhibition balance is biased towards the latter; (iii) force hyperpolarized thalamocortical neurons to fire bursts of action potentials back to the cortex and then fall into a prolonged silence, when these cells are in a bursting mode. In fact, TMS perturbations could reveal the presence of adaptation mechanisms and of the ensuing OFF-periods in all patients, regardless of their background EEG pattern (Supplementary Table 2), of the prevalence of spontaneously occurring slow waves (Supplementary Fig. 3), and of pre-stimulus ongoing activity (Fig. 2B, C and D).”

Intriguingly, the strength of adaptation mechanisms has been considered as a key factor in shaping the behavior of dynamical systems that are employed to model sleep-like activity (Compte et al., J Neurophysiol. 2003 May;89(5):2707-25; Tartaglia and Brunel, Sci Rep. 2017 Sep 20;7(1):11916; Mattia and Sanchez-Vives, Cogn Neurodyn. 2012 Jun;6(3):239-50). In future studies, it would be very interesting to explore the formal relationships between the notion of bistability as a system-wide phenomenon defined in the framework of dynamical systems (Freyer et al., J Neurosci. 2009 Jul 1;29(26):8512-24; Freyer et al., J Neurosci. 2011 Apr 27;31(17):6353-61; Jercog et al., Elife. 2017 Aug 4;6) and cortical bistability as experimentally observed here.”

2. I couldn't follow how the correlation analysis of Figure 3 supports the causality argument implicit in the phrase, “disrupt local causality”. For a start, are the correlations within or across participants? (in [1] it seems they are within single subjects, here there is just one regression for each pair of variables). A within-subject analysis would be more convincing – i.e. in those single trials, when there is a slow wave, there is no burst, and

conversely, there is a burst (even if it is diminished) if there is no slow wave. I am not convinced that a causal process is established if the regression is across participants.

Fig. 3 shows the relationships between the presence of the slow wave, the occurrence of an OFF-period and the disruption of local causality. Here, local causality is explicitly measured by the Phase Locking Factor (PLF), which quantifies the duration of the deterministic effects of an initial neuronal activation (in this case triggered TMS). The correlation between the timing of the OFF-period and the timing of the drop of PLF is performed to parallel - at the macroscale, coarse level of the EEG recordings - the results of *in vitro* (Sanchez-Vives and Mc Cormick, Nat Neurosci. 2000 Oct;3(10):1027-34) and *in vivo* (Timofeev et al., Cereb Cortex. 2000 Dec;10(12):1185-99) observations suggesting that the resumption of cortical activity following an OFF-period is a stochastic process.

Regarding the correlation analyses, the PLF cannot be computed for single trials since it is calculated the absolute value of the average of the Hilbert Transform across trials. As such, it measures the coherence of the response to a perturbation across trials, and can be used to quantify the duration of the deterministic effect of a given input.

In a previous mesoscale intracranial work (Pigorini et al., Neuroimage. 2015 May 15;112:105-113) the correlation between the timing of OFF-periods and the duration of PLF was performed, for each subject, across contacts (contacts showing an earlier OFF-period also showed a shorter PLF duration). When using scalp EEG recordings, as in the present study, this kind of analysis across contacts is not feasible due to the intrinsic limitations represented by volume conduction. For this reason, we show a similar correlation across subjects (i.e. in patients where the OFF-period occur earlier also the PLF drops earlier).

We agree with the Reviewer that the correlation between the three relevant measures (the amplitude of the TMS-evoked slow wave, the timing of the OFF-period and the duration of causal effects) would be best performed at the single-subject level. In the case of scalp recordings in patients this can ideally be done by performing longitudinal measurements. To this regard we think that the special case of the patient recovering consciousness (Patient 16; Fig. 5) is particularly relevant: indeed, within this single patient, a progressive disappearance of the OFF-period was followed by the recovery of local causal effects and overall complexity.

3. The paper claims that the response is “regardless of stimulation site” although I only see BA6 and BA7 in the corresponding Figure 2A.

Indeed, the cortex of UWS patients was mapped in a systematic but not in an extensive manner: in each patient, we aimed at targeting both the frontal and the parietal lobe bilaterally. In some patients these targets were not accessible due to lesions and extracranial derivations for liquor drainage (see also response to Reviewer #2).

Thus, a correct description of our results is that OFF-periods were always elicited in any of the targeted sites (precise stimulation location is now included in Supplementary Table 2 in the revised version of the manuscript). We agree with the Reviewer that “irrespective of stimulation site” is too strong of a statement. For this reason we have changed the title of Fig. 2 as follows:

“TMS evokes a stereotypical sleep-like OFF-period at all targeted sites and irrespective of pre-stimulus activity”

4. Discussion: Bistability, if present, has its neurobiological moderators/correlates but it is ultimately a system-wide phenomenon and I thought the Discussion might engage in such considerations.

In the present work, we attempt to draw a link between a neurobiological mechanism (neuronal adaptation and the ensuing OFF-periods) and system-wide dynamics as assessed experimentally by PCI. Although very interesting, formally describing cortical bistability as a system-wide phenomenon in dynamical systems is out of the scope of the present study (but, see our response to point 1). Nonetheless, to acknowledge the importance of this perspective, following the Reviewer's suggestion, we included a paragraph in the revised version of the Discussion section that reads (Discussion - Page 12):

“Intriguingly, the strength of adaptation mechanisms has been considered as a key factor in shaping the behavior of dynamical systems that are employed to model sleep-like activity 15,55,56. In future studies, it would be crucial to explore the formal relationships between cortical bistability as experimentally observed here and the notion of bistability as a system-wide phenomenon defined in the framework of dynamical systems 57–59.”

Minor:

1. The text jumps from Figure 1A, to 2A and 2B, then back to the rest of Figure 1, then to Figure 3, without ever referring to 2C or 2D. It makes for a confusing read and I wonder if the authors could try for a more linear exposition of their panels in the main text.

We now included the proper reference to Fig. 2B, 2C and 2D in the revised version of the Results section thus referring to all panels pertaining to Fig. 2 before reporting the results included in Fig. 3. We hope these changes (together with all text changes suggested also by this and the other Reviewers) improved the overall readability of the revised version of manuscript.

2. Please provide a p-value for PCI (p7).

We now indicate this information in the revised version of the Results section.

References:

- 1. A. Pigorini, S. Sarasso, P. Proserpio, C. Szymanski, G. Arnulfo, S. Casarotto, M. Fecchio, M. Rosanova, M. Mariotti, G. Lo Russo, J. M. Palva, L. Nobili, M. Massimini, Bistability breaks-off deterministic responses to intracortical stimulation during non-REM sleep, *Neuroimage* 112, 105–113(2015).**
- 2. M. V. Sanchez-Vives, M. Massimini, M. Mattia, Shaping the Default Activity Pattern of the Cortical Network, *Neuron* 94, 993–1001 (2017).**
- 3. Freyer F, Roberts JA, Becker R, Robinson PA, Ritter P, Breakspear M (2012). A canonical model of multistability and scale-invariance in biological systems. *PLoS Computational Biology* 8: e1002634**
- 4. Freyer F, Roberts JA, Becker R, Robinson PA, Ritter P, Breakspear M (2011). Dynamic mechanisms of multistability in the human alpha rhythm. *Journal of Neuroscience* 31: 6353-6361.**

REVIEWERS' COMMENTS:

Reviewer #1 (Remarks to the Author):

The revised manuscript by Rosanova and colleagues is an interesting study of unresponsive wakefulness syndrome (UWS) that provide electrographic evidence for an abnormal sleep-like bi-stability that could explain the loss of consciousness in patients with UWS. The authors submitted thoughtful and thorough responses to this reviewer's comments. Revisions to the original manuscript now make clear several aspects of the methods and analysis. Additional data, including supplementary figures, support the study's conclusion. Lastly, revisions were made to improve the readability of results and discussion.

Reviewer #2 (Remarks to the Author):

The author answer correctly and in extensive way to the questions of the reviewer .

The comments requested for introduction, methods and discussion were clearly explained.

The argument is relatively novel for the scientific community mainly in the methods.

The supplements reported by the authors helped to understand the results and answer to some comments.

The paper is well written and the figures are clear.

Reviewer #3 (Remarks to the Author):

The authors' responses to me concerns are satisfactory.

I still feel, for a broad readership like Nature Communications, "cortical bistability" is going to be more broadly understood as a system with two distinct states, and not in the restricted sense used in sleep physiology. But I take the authors' point and think their revisions on this point are reasonable (although its not that hard to actually show bimodal spectra/cross-spectral fluctuations). I look forward to the authors next steps in this direction, as outlined in their response letter.